# Deep Neural Networks Can Learn Generalizable Same-Different Visual Relations

## Abstract

Although deep neural networks can achieve human-level performance on many object recognition benchmarks, prior work suggests that these same models fail to learn simple abstract relations, such as determining whether two objects are the *same* or *different*. Much of this prior work focuses on training convolutional neural networks to classify images of two same or two different abstract shapes, testing generalization on within-distribution stimuli. In this article, we comprehensively study whether deep neural networks can acquire and generalize same-different relations both within and out-of-distribution using a variety of architectures, forms of pretraining, and fine-tuning datasets. We find that certain pretrained transformers can learn a same-different relation that generalizes with near perfect accuracy to out-of-distribution stimuli. Furthermore, we find that fine-tuning on abstract shapes that lack texture or color provides the strongest out-of-distribution generalization. Our results suggest that, with the right approach, deep neural networks can learn generalizable same-different visual relations.

## 1 Introduction

Humans and a wide variety of non-human animals can easily recognize whether two objects are the same as each other or whether they are different (see Figure 1; Martinho III & Kacelnik, 2016; Christie, 2021; Gentner et al., 2021; Hespos et al., 2021). The abstract concept of equality is simple—even 3-month-old infants (Anderson et al., 2018) and honeybees (Giurfa, 2021) can learn to distinguish between displays of two same or two different objects. Some researchers have even argued that it serves amongst a number of other basic logical operations as a foundation for higher-order cognition and reasoning (Gentner & Goldin-Meadow, 2003; Gentner & Hoyos, 2017). However, in contrast to humans and animals, recent work has argued that deep neural networks struggle to learn this simple relation (Ellis et al., 2015; Gülçehre & Bengio, 2016; Stabinger et al., 2016; Kim et al., 2018; Webb et al., 2020; Puebla & Bowers, 2022). This difficulty is surprising given that deep neural networks achieve human or superhuman performance on a wide range of seemingly more complex visual tasks, such as image classification (Krizhevsky et al., 2012; He et al., 2016), segmentation (Long et al., 2015), and generation (Ramesh et al., 2022).

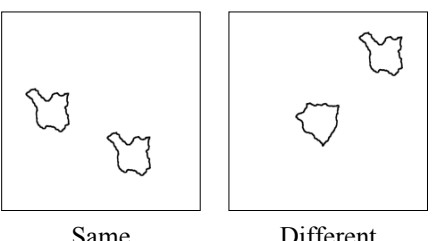

Same   Different

Figure 1: **Same or different?** For humans and a number of animal species, it is trivial to recognize that the image on the left contains two of the same objects, while the image on the right contains two different objects. Surprisingly, prior research has suggested that deep neural networks struggle to learn to discriminate between these images.

Past attempts to evaluate same-different relations in neural networks have generally used the following methodology. Models are trained to classify images containing either two of the same or two different abstract objects, such as those in Figure 1. A model is considered successful if it is then able to generalize the same-different relation to unseen shapes after training. Convolutional neural networks (CNNs) trained from scratch fail to learn a generalizable relation, and tend to memorize training examples (Kim et al., 2018; Webb et al., 2020). However, deep neural networks have been shown to

successfully generalize the same-different relation in certain contexts. This generalization is either limited to in-domain test stimuli (Funke et al., 2021; Puebla & Bowers, 2022) or requires architectural modifications that build in an inductive bias towards relational tasks at the expense of other visual tasks (Kim et al., 2018; Webb et al., 2020; 2023a;b; Kerg et al., 2022; Geiger et al., 2023; Altabaa et al., 2023). Given these limited successes, an open question remains: without architectural modifications that restrict model expressivity in general, can standard neural networks learn an abstract same-different relation that generalizes to both in- and out-of-distribution stimuli?

Addressing this question requires going beyond past work in a number of ways. First, most previous studies test for in-distribution generalization—that is, they use test stimuli that are visually similar to the training stimuli. We believe that out-of-distribution generalization provides much stronger evidence that a model has learned a genuine abstract relation without relying on spurious features. Second, the existing literature uses training stimuli that demonstrate the same-different relation with either closed curves (as in Figure 1) or simple geometric shapes. It is unclear whether training on these types of objects is the most helpful for learning the relation versus more naturalistic objects that more closely resemble data seen during pretraining. Finally, most prior work focuses on convolutional architectures, but Vision Transformers (ViTs) (Dosovitskiy et al., 2020) adapted from the language domain (Vaswani et al., 2017) have recently emerged as a competitive alternative to CNNs on visual tasks. Self-attention, a key feature of ViTs, may provide an advantage when learning abstract visual relations—indeed, the ability to attend to and relate any part of a stimulus to any other part may be crucial for relational abstraction.

In this article, we address these limitations and comprehensively investigate how neural networks learn and generalize the same-different relation from image data. Our main findings are as follows:

- Fine-tuning pretrained ResNet and ViT models on the same-different relation enables both architectures to generalize the relation to unseen objects in the same distribution as the fine-tuning set. In particular, CLIP pretraining results in nearly 100% in-distribution test accuracy for ViT models, and close to that for ResNet models. (Section 3.1)

- Under certain conditions, CLIP-pretrained ViTs can reliably generalize the same-different relation to out-of-distribution stimuli with nearly 100% accuracy (Section 3.2). Furthermore, these models can transfer the relation with up to 90% test accuracy to a photorealistic same-different dataset of 3D objects *without any fine-tuning on the 3D setting* (Section 3.3). These results suggest that these models acquire a generalizable abstract concept of equality.

- Different fine-tuning datasets lead to qualitatively different patterns of generalization—fine-tuning on more visually abstract objects (which do not contain color or texture) results in stronger out-of-distribution generalization, whereas fine-tuning on more naturalistic objects fails to generalize. (Section 3.2)

- ViTs generally prefer to determine equality between objects by comparing their color or texture, only learning to compare shape when the fine-tuning dataset lacks color and texture information. However, we find that CLIP pretraining helps to mitigate this preference for color and texture. (Section 4)

- We find CLIP ViTs can achieve strong generalization via computing a "fuzzy" same-different relation, using a threshold of embedding similarity to judge "same" vs. "different." While our results show perfect OOD generalization for the 4 datasets used for fine-tuning, testing on 13 additional same-different datasets from Puebla & Bowers (2022) reveals that models can struggle to generalize to some of these additional datasets in which objects are much more visually similar to each other than the objects in the model's fine-tuning data. (Appendix A.1)

## 2   Methods

We operationalize the same-different task consistently with prior work, e.g. Fleuret et al. (2011). Models are asked to perform a binary classification task on images containing either two of the same objects or two different objects (see the second and third rows of Figure 2). Models are either trained from scratch or fine-tuned on a version of this task with a particular type of stimuli (see Section 2.1 below). After training

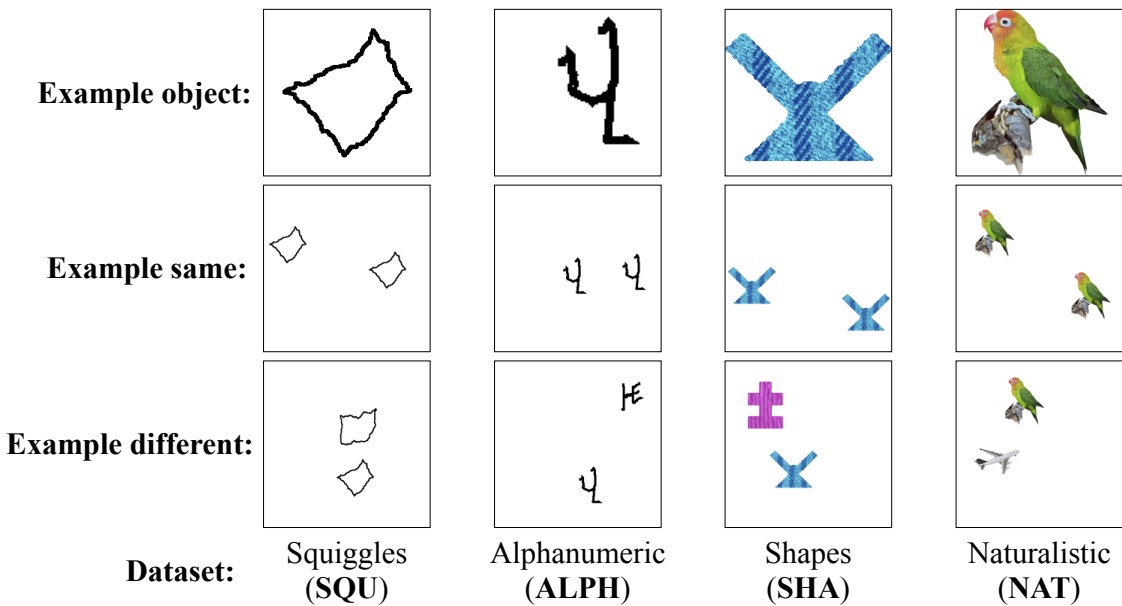

Figure 2: **Example stimuli from all four datasets.** Each column represents one of the four same-versus-different datasets as indicated by the label beneath the stimuli. The top row shows an example object that is used to form the stimuli that comprise each dataset, while the second and third rows show an example "same" vs. "different" stimulus, respectively.

or fine-tuning, model weights are frozen, and validation and test accuracy scores are computed on sets of same-versus-different stimuli containing unfamiliar objects. These can be either be the same type of objects that they were trained or fine-tuned on (in-distribution generalization) or different types of objects (out-of-distribution generalization). Thus, in order to attain high validation and test accuracy scores, the model must successfully generalize the learned same-different relation to novel objects. This type of generalization is more challenging than the standard image classification setting because of the abstract nature of what defines the classes—models must learn to attend to the relationship between two objects rather than learn to attend to any particular visual features of those objects in the training data.

## 2.1 Training and Evaluation Datasets

We construct four same-versus-different datasets using four different types of objects (see Figure 2) ranging from abstract shapes to naturalistic images that are more familiar to pretrained models. We use the following objects to create these four datasets:

1. **Squiggles (SQU).** Randomly generated closed shapes following Fleuret et al. (2011).[1] Most studies in the machine learning literature on the same-different relation uses this dataset (Kim et al., 2018; Funke et al., 2021; Puebla & Bowers, 2022; Messina et al., 2022).

2. **Alphanumeric (ALPH).** Sampled handwritten characters from the Omniglot dataset (Lake et al., 2015).

3. **Shapes (SHA).** Textured and colored shapes from Tartaglini et al. (2022). Objects that match in shape, texture, and color are considered the same, whereas objects that differ along all three dimensions are considered different.

---

[1]The original method from Fleuret et al. (2011) produces closed contours with lines that are only one pixel thick. For our chosen image and object size, these shapes become very difficult to see. We correct this by using a dilation algorithm to darken and thicken the lines to a width of three pixels.

4. **Naturalistic (NAT).** Photographs of real objects on white backgrounds from Brady et al. (2008). These stimuli are the most similar to the data that the pretrained models see before fine-tuning on the same-different task.

Each stimulus is an image that contains two objects that are either the same or different. We select a total of 1,600 unique objects for each dataset. These objects are split into disjoint sets of 1,200, 300, and 100 to form the training, validation, and test sets respectively. Unless otherwise specified, the training, validation, and test sets each contain 6,400 stimuli: 3,200 same and 3,200 different. To construct a given dataset, we first generate all possible pairs of same or different objects—we consider two objects to be the same if they are the same on a pixel level.[2] Next, we randomly select a subset of the possible object pairs to create the stimuli such that each unique object is in at least one pair. Each object is resized to 64x64 pixels, and then a pair of these objects is placed over a 224x224 pixel white background in randomly selected, non-overlapping positions. We consider two objects in a specific placement as one unique stimulus—in other words, a given pair of objects may appear in multiple images but in different positions (but with all placements of the same two objects being confined to either the training, validation, or test set). All object pairs appear the same number of times to ensure that each unique object is equally represented.

## 2.2 Models and Training Details

We evaluate one convolutional architecture, **ResNet-50** (He et al., 2016), and one Transformer architecture, **ViT-B/16** (Dosovitskiy et al., 2020). We also evaluate three pretraining procedures: (1) **randomly initialized**, in which all model parameters are randomly initialized (Kaiming normal for ResNet-50 and truncated normal for ViT-B/16) and models are trained from scratch, (2) **ImageNet**, in which models are pretrained in a supervised fashion on a large corpus of images (ImageNet-1k for ResNet-50 and ImageNet-21k for ViT-B/16; Deng et al., 2009) with category labels such as "barn owl" or "airplane," and (3) **CLIP** (Radford et al., 2021), in which models learn an image-text contrastive objective where the cosine similarity between an image embedding and its matching natural language caption embedding is maximized. Unlike ImageNet labels, CLIP captions contain additional linguistic information beyond category information (e.g. "a photo of a barn owl in flight"). To address the difference in parameter count between ResNet-50 and ViT-B/16 (23M versus 86M parameters), we also provide results for ImageNet-pretrained ConvNeXt-B Liu et al. (2022) and DeiT-S Touvron et al. (2021) in Appendix A.4 (89M and 22M parameters respectively).

We adapt all models to the same-different task by appending a linear classifier to the output of the visual backbone. For models not trained from scratch, we directly fine-tune on training sets from Section 2.1. Each model is trained from scratch or fine-tuned for 70 epochs with a batch size of 128, updating all parameters. We use a binary cross-entropy loss. For each architecture and pretraining combination, we perform hyperparameter tuning via grid search over the initial learning rate (1e-4, 1e-5, 1e-6, 1e-7, 1e-8), learning rate scheduler (`exponential`, `ReduceLROnPlateau`), and optimizer (`SGD`, `Adam`, `AdamW`). We select the best performing training configuration from the grid search according to in-distribution validation accuracy, and then train a model with those hyperparameters five times with different random seeds. We report the median test results across those five seeds.

# 3 Generalization to Unseen Objects

## 3.1 In-Distribution Generalization

We first measure the performance of each model on test data containing the same types of objects used to train or fine-tune the model; e.g. models fine-tuned on pairs of handwritten characters are then tested on handwritten characters that were not seen during training. We refer to this as the in-distribution performance of the model. The starred (∗) result in Figure 3 shows the in-distribution median test accuracy of randomly-initialized ResNet-50 models trained on the Squiggles dataset, which contains the same type of closed contours

---

[2]There is some ambiguity in how to define sameness. For example, one could imagine a same-different task in which two objects drawn from the same category are considered the same, such as two different images of the same species of parrot. Furthermore, two objects can be the same in some dimensions but differ in others (see Section 4.2). Unless otherwise stated, we take "same" to mean "exactly the same."

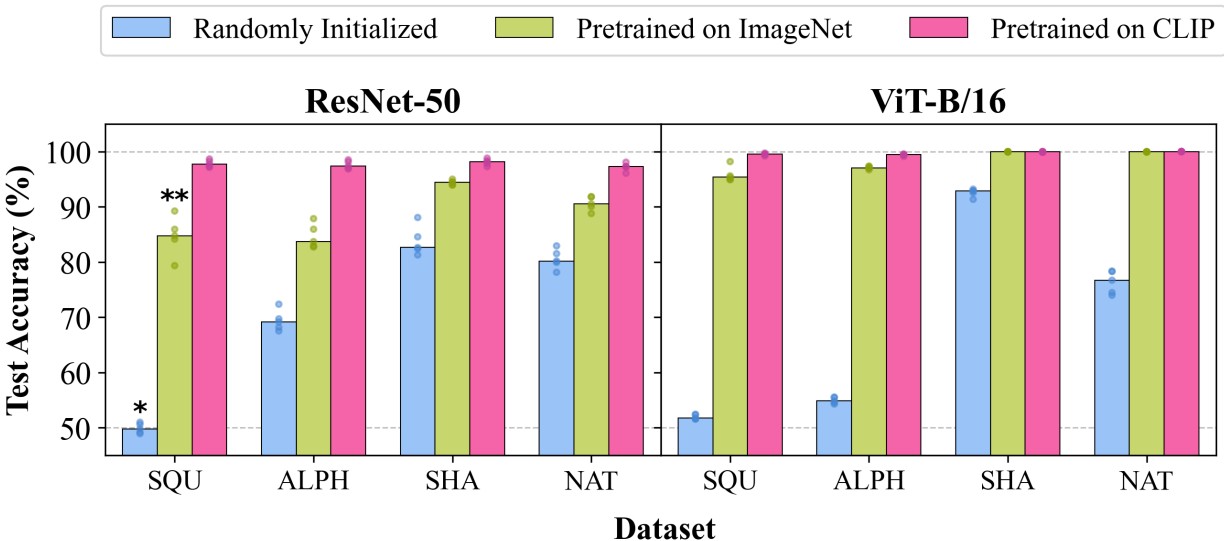

Figure 3: **In-distribution test accuracy by architecture and pretraining method.** Bars show median accuracy over 5 runs (individual points also shown), with the bar color denoting pretraining type and the x-axis denoting the dataset used for fine-tuning. See Section 2.1 for dataset descriptions, Section 2.2 for model details, and Figure 2 for visual examples. The **starred** (∗) result is a replication of findings from prior work showing that CNNs trained from scratch on stimuli like the images in Figure 1 attain chance-level test accuracy. The **double-starred** (∗∗) result mirrors Funke et al. (2021) and Puebla & Bowers (2022), who show that ImageNet-pretrained CNNs attain substantially higher in-distribution test accuracy relative to the same architectures trained from scratch.

used by much of the prior work on the same-different relation (Fleuret et al., 2011; Kim et al., 2018; Funke et al., 2021; Puebla & Bowers, 2022; Messina et al., 2022). Confirming the primary findings from prior work, these models do not attain above chance level test accuracy. The same pattern holds for randomly initialized ViT-B/16 models.

However, as the rest of Figure 3 shows, pretrained models exhibit substantially improved in-distribution accuracy compared to randomly initialized models across all four datasets. In particular, models pretrained with CLIP demonstrate the largest improvements, attaining nearly 100% test accuracy irrespective of fine-tuning dataset. Even without any fine-tuning, CLIP features appear to be highly useful for the same-different task—linear probes trained to do the same-different task using CLIP ViT-B/16 embeddings of stimuli without any fine-tuning achieve between 80% and 100% median in-distribution test accuracy depending on the dataset (Table 13, Appendix A.7). Differences in performance can also be observed between architectures, with ViT-B/16 models consistently outperforming ResNet-50 after pretraining.[3] These differences are likely not a result of a difference in parameter counts, as a similar gap in performance can be observed between ImageNet ConvNeXt-B and ImageNet ViT-B/16, despite ConvNeXt-B being slightly larger (Appendix A.4).

Another main finding is that the two visually abstract, shape-based datasets (SQU and ALPH) appear to pose more of a challenge to models than the SHA and NAT datasets—models attain noticeably higher in-distribution accuracy on the latter two across architectures and pretraining methods (although the effect is small for CLIP-pretrained models). This difference may be due to the color and texture information that is available in these datasets, which provides additional dimensions over which objects can be compared. We explore the possibility that some models find it easier to evaluate equality using color or texture in addition to or instead of shape information in Section 4.

---

[3]ViTs also demonstrate qualitatively different training dynamics compared to CNNs, appearing to generalize the same-different relation within the first few epochs of training. Furthermore, ViTs learn more *smoothly* than ResNets. See Appendix A.2 for figures of training and accuracy curves.

Table 1: **Out-of-distribution (OOD) test accuracy for CLIP models fine-tuned on each dataset.** Rows indicate the dataset that models are fine-tuned on, while columns indicate the test dataset. Each cell is the median performance over five random seeds. The rightmost column labeled "Avg." is the row-wise average of accuracy scores across OOD test sets (i.e. off-diagonal values), which indicates how well a model fine-tuned on a given dataset is able to generalize to other datasets. The bottom row labeled "Avg." is the column-wise average across off-diagonal values, indicating how difficult it is for models fine-tuned on other datasets to generalize to the given dataset. Note that the **bolded** diagonals are the pink bars in Figure 3. OOD generalization results for all models are in Appendix A.3; Appendix A.5 shows median AUC-ROC scores.

| CLIP ResNet-50 | | | | | | CLIP ViT-B/16 | | | | | |
|---|---|---|---|---|---|---|---|---|---|---|---|
| | $\leftarrow$ **Test** $\rightarrow$ | | | | | | $\leftarrow$ **Test** $\rightarrow$ | | | | |
| **Train** ↓ | SQU | ALPH | SHA | NAT | Avg. | **Train** ↓ | SQU | ALPH | SHA | NAT | Avg. |
| SQU | **97.7** | 82.9 | 86.9 | 82.0 | 83.9 | SQU | **99.6** | 97.7 | 99.1 | 96.7 | 97.8 |
| ALPH | 82.1 | **97.4** | 92.8 | 91.8 | 88.9 | ALPH | 55.3 | **99.4** | 99.6 | 91.2 | 82.0 |
| SHA | 56.0 | 78.1 | **98.1** | 96.1 | 76.7 | SHA | 50.0 | 55.4 | **100** | 100 | 68.5 |
| NAT | 50.1 | 59.3 | 93.4 | **97.3** | 67.6 | NAT | 50.0 | 68.0 | 99.8 | **100** | 72.6 |
| Avg. | 62.7 | 73.4 | 91.1 | 90.0 | | Avg. | 51.8 | 73.7 | 99.5 | 95.9 | |

## 3.2 Out-of-Distribution Generalization

The previous section showed that pretrained models can generalize to unseen, in-distribution objects. However, if a model learns a truly abstract notion of same-different, it should be able to generalize the same-different relation to any two objects regardless of their particular visual features. Thus, model performance on stimuli that are substantially different from training stimuli is a stronger measure of abstraction. We therefore measure test accuracy for each model across all four datasets, yielding one in-distribution score and three out-of-distribution (OOD) scores per model. Table 1 shows median test accuracy over five seeds for CLIP-pretrained models; full generalization tables for all pretraining styles and architectures can be found in Appendix A.3.

Overall, CLIP ViT-B/16 models fine-tuned on the Squiggles task exhibit the strongest OOD generalization, achieving >95% median test accuracy on the three out-of-distribution datasets.[4] As in the previous section, models fine-tuned on objects with visually abstract shape features only (SQU and ALPH) behave differently than those fine-tuned on datasets containing objects with shape, color, and texture features (SHA and NAT). The SQU and ALPH models generally attain high OOD test accuracy. On the other hand, models fine-tuned on the SHA or NAT datasets generalize well to each other but struggle to generalize to the SQU and ALPH tasks. Note that some of this effect can be attributed to miscalibrated bias, but not the entire effect—see Appendix A.5 for details.

Another way to understand the generalization pattern in Table 1 is that the more "challenging" a dataset is to generalize the same-different relation to, the more effective it is as a fine-tuning dataset for inducing out-of-distribution generalization. For example, CLIP ViT-B/16 models fine-tuned on datasets other than Squiggles attain a median test accuracy of only 51.8% on the Squiggles task on average, whereas CLIP ViT-B/16 fine-tuned on Squiggles attains an average OOD test accuracy of 97.8%. On the other hand, the Shapes dataset is easy for models fine-tuned on other datasets to generalize to (99.5% accuracy on average), but CLIP ViT fine-tuned on that "easier"

Table 2: **Average pairwise cosine similarity between CLIP embeddings of training stimuli within each dataset.** Because $n = 6,400$ for each dataset, averages are computed over approximately 20M pairs. We extract CLIP embeddings *before* fine-tuning on the same-different task. For similarities afterwards, see Appendix A.8.

| Dataset ↓ | **ResNet-50** | **ViT-B/16** |
|---|---|---|
| noise | 0.992 | 0.993 |
| SQU | 0.929 | 0.940 |
| ALPH | 0.881 | 0.889 |
| SHA | 0.855 | 0.801 |
| NAT | 0.788 | 0.805 |

---

[4]It is worth noting that both this model and CLIP ResNet-50 fine-tuned on the ALPH task (the model with the second best OOD generalization performance) exhibit some degree of sensitivity to the random seed used during fine-tuning. Most random seeds result in nearly 100% OOD generalization for ViT or >80% for ResNet across all datasets, while some seeds result in substantially lower performance (1/5 seeds for ViT and 2/5 for ResNet). No other model configurations exhibit this bimodal behavior. See Appendix A.10 for details.

dataset attains an average OOD test accuracy of only 68.5%. This pattern of Squiggles being more "difficult" to generalize to persists across architectures and pretraining methods (Appendix A.3).

We further study why fine-tuning on different datasets results in different generalization behaviors by computing the average cosine similarity between objects in a given dataset using pretrained CLIP embeddings (Table 2). This value provides information about the visual variation in each dataset through the lens of a specific model: a higher number means that stimuli in that dataset are generally embedded more closely together by that model. Before fine-tuning, pretrained CLIP models embed Squiggles stimuli more closely together than stimuli from other datasets, potentially explaining the difficulty of that dataset as a test of OOD generalization. We also note what seems to be a correlation between "closeness" of stimuli in a model's embedding space and ability of models fine-tuned on that dataset to generalize OOD. Given that random noise is embedded even more closely together than Squiggles stimuli, we fine-tune models on the same-different task comparing patches of random noise and measure their OOD generalization in Appendix A.9. We find that models fine-tuned on noise exhibit weaker OOD generalization than models fine-tuned on Squiggles, indicating that "closeness" of stimuli is not a perfect correlate of OOD generalization and is only part of the story (see Section 4).

### 3.3 Out-of-Distribution Generalization to Photorealistic Stimuli

In the previous section, we demonstrated that CLIP-pretrained models fine-tuned on one set of same-different stimuli can consistently generalize the relation to other visually distinct sets of stimuli. Despite these successes, one potential criticism is that the types of artificial stimuli we use—albeit having proved significantly challenging in previous attempts to solve the same-different task—lack the additional complexities involved in recognizing abstract visual relations in real-world environments. In particular, following prior work, the objects in our "same" stimuli are the same as each other at the pixel level. It is possible that models in this conventional setting learn to generalize the same-different relation by simply recognizing whether small patches of pixels or even single pixels have the same values rather than comparing whole objects. Relying on pixel-level mechanisms to adjudicate between same and different would fail in photorealistic settings, since two instances of the same 3D object can differ at the pixel level due to differences in lighting, rotation, and depth of field. Given the successful OOD generalization of our fine-tuned models in an artificial setting, we wanted to test whether our findings extend to more challenging real-world settings.

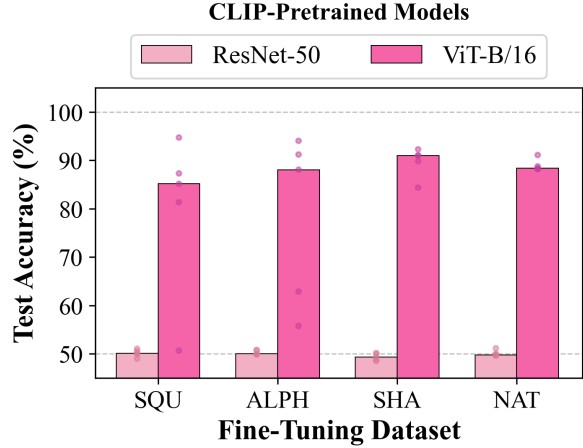

Figure 4: **Median photorealistic test accuracy for CLIP models fine-tuned on SQU, ALPH, SHA, & NAT.** Models are tested on a realistic same-different dataset (see Figure 5) without any fine-tuning on this dataset. Test accuracies and AUC-ROC for all models can be found in Appendix A.11.

To this end, we evaluate the fine-tuned models from Sections 3.1 and 3.2 on a dataset of 1,024 photorealistic same-different stimuli that we generated (see Figure 5). Each stimulus is a 224x224 pixel image depicting a pair of same or different 3D objects arranged on the surface of a table in a sunlit room. We created these images in Blender, a sophisticated 3D modeling tool, using a set of 16 unique 3D models of different objects that vary in shape, texture and color. To construct the dataset, we first generate all possible pairs of same or different objects, then select a subset of the possible "different" pairs such that each object appears in two pairs. This ensures that all objects are equally represented and that an equal number of "same" and "different" stimuli are created. We create 32 unique stimuli for each pair of objects by placing them on the table in eight random configurations within the view of four different camera angles, allowing partial

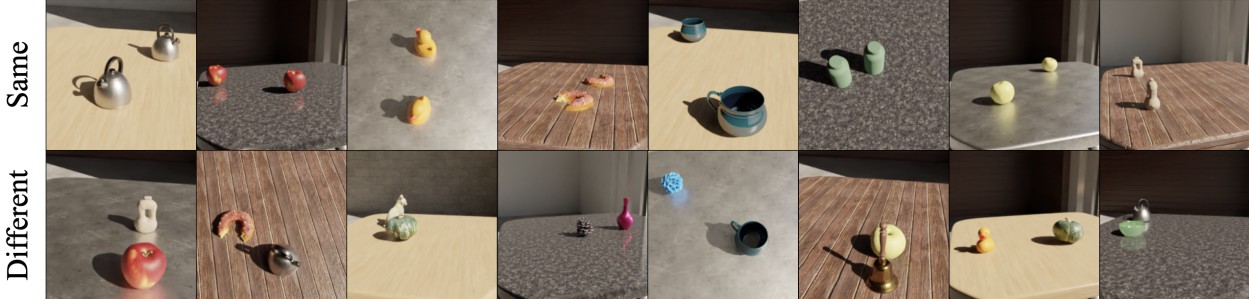

Figure 5: **Examples of "same" and "different" photorealistic stimuli.** The textures of the table surface and background wall are randomly selected from a set of four options each. No two objects in an image are the same on a pixel level. See Appendix A.11 for images of all 16 3D objects.

occlusions. Each individual object is also randomly rotated around its Z-axis in each image—because 11 of the objects lack rotational symmetry, these rotations provide an additional challenge, especially for "same" classifications.[5]

We evaluate the SQU, ALPH, SHA, and NAT fine-tuned models from the previous sections on the photorealistic dataset *without any additional fine-tuning.* Figure 4 shows median test accuracy on the photorealistic dataset for CLIP-pretrained models across the same five random seeds reported in Sections 3.1 and 3.2. Surprisingly, CLIP-pretrained ViT models generalize with 80-90% median test accuracy to the photorealistic same-different stimuli despite only receiving fine-tuning on pixel-level sameness between 2D objects, indicating that their robust generalization of the same-different relation is not limited to our particular definition of the same-different task. On the other hand, all other pretraining and architecture combinations including CLIP-pretrained ResNets fail to generalize consistently to the photorealistic stimuli (see Appendix A.11). These results suggest that, with careful choices of architecture and pretraining, fine-tuning on simplistic 2D stimuli may be sufficient for learning an abstract same-different relation that generalizes to 3D objects despite the additional visual complexities of real-world settings.

## 4 Examination of Inductive Biases

What features are these models actually using to decide whether two objects in an image are the same? Because we train models without explicit guidance as to how to solve the task, we expect that any inductive biases a given model may have will influence how it learns the same-different relation. Previous work has claimed that CNN models trained on ImageNet are often biased towards texture over shape (Geirhos et al., 2019; Hermann et al., 2020). This may be related to results from Kim et al. (2018) that show poor performance for CNNs trained from scratch on textureless shapes. In this section, we investigate whether and how these inductive biases influence model behavior for the visual same-different task.

### 4.1 Grayscale and Graymasked Objects

We train models on one of three variants of the Shapes dataset: objects are either kept the same (Figure 6a, "Color"), grayscaled to preserve texture but remove color (Figure 6a, "Grayscale"), or completely covered in gray to remove both texture and color (Figure 6a, "Masked"). If a model is biased towards color, we would expect performance to drop on the Grayscale and Masked datasets, and if it is biased towards texture, we would expect performance to suffer on the Masked dataset. In this setup, only a model with an understanding of shape would generalize effectively to all three settings. We train/fine-tune on each of these three variants for randomly initialized, ImageNet-pretrained, and CLIP-pretrained models.

---

[5]We also test models on a version of the photorealistic dataset where "same" objects are always rotated identically. We find that performance for most models improves, albeit very slightly; see Appendix A.11.

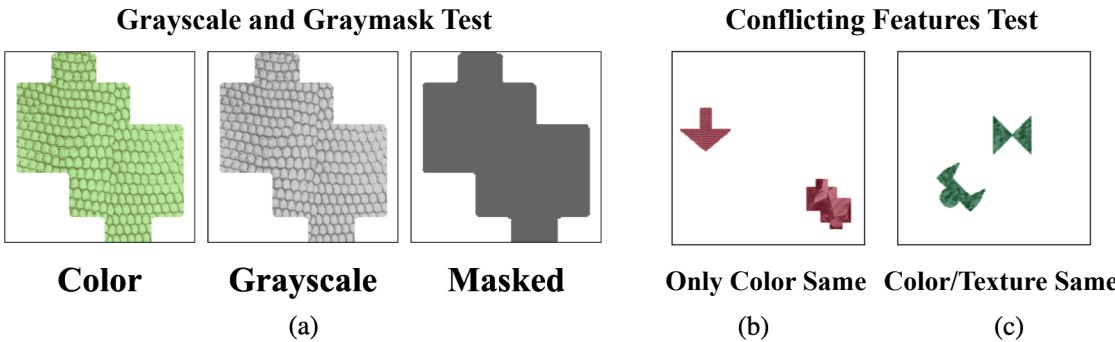

Figure 6: **Examples of stimuli used to test inductive biases.** Figure (a) shows examples of objects from the three versions of the Shapes dataset used to produce results in Figure 7. Figures (b) and (c) are examples of stimuli with conflicting signals used in Section 4.2, where either color is the same while texture and shape are different, or color and texture are the same while shape is different.

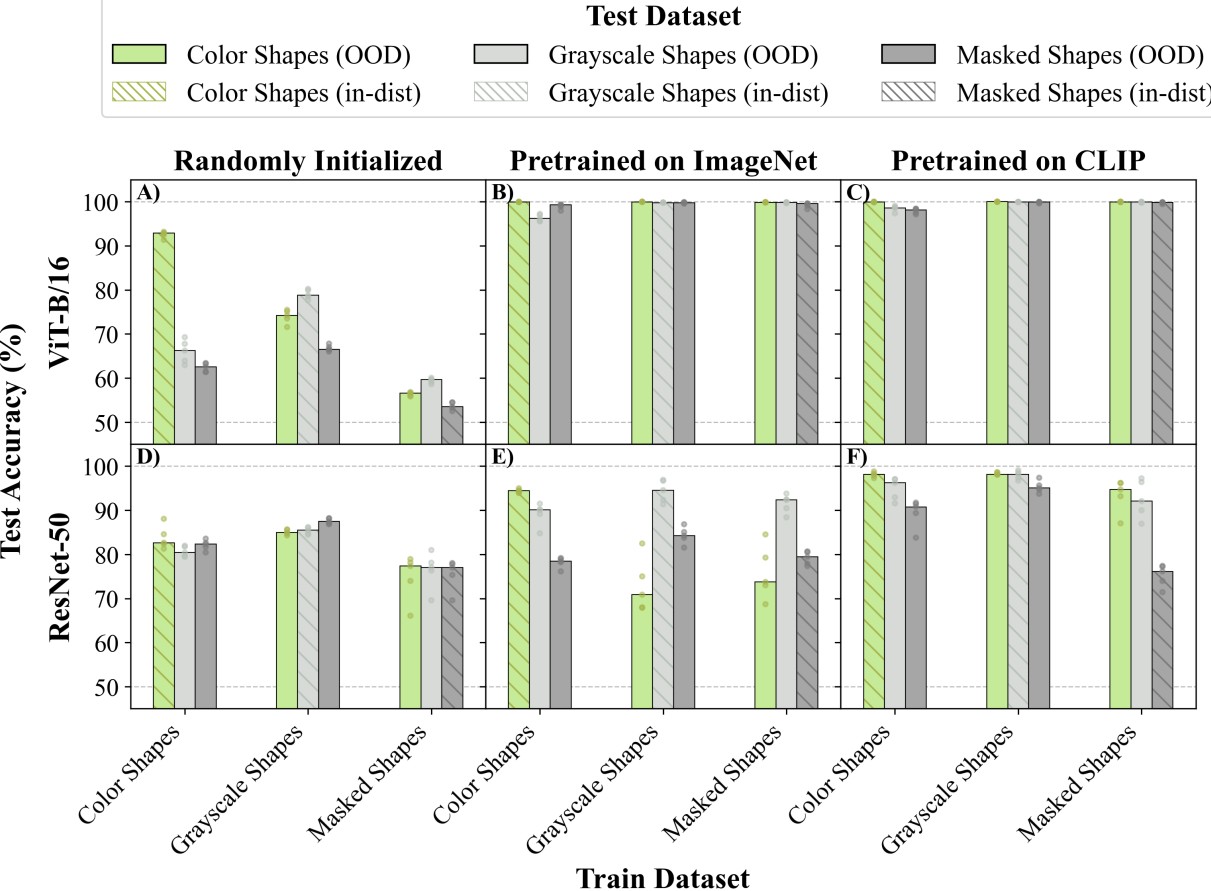

Figure 7: **Test accuracy for models trained or fine-tuned on one version of the Shapes dataset (Color, Grayscale, Masked) and then tested on all three versions of the dataset.** Example stimuli are shown in Figure 6a. Hatched bars indicate in-distribution accuracy. Median results for the same hyperparameters trained for five different seeds are reported, with individual runs also plotted as translucent points.

Figure 7A shows the test performance of randomly initialized ViT-B/16 trained on either Color, Grayscale, or Masked versions of the Shapes dataset (Figure 6a) and then tested on novel objects from each of those distributions. First, we can look at in-distribution generalization accuracy, represented by the hatched bars. From Figure 7A, we can see that ViT-B/16 trained from scratch can only achieve high in-distribution accuracy for the Color Shapes dataset (92.9%); the hatched gray and dark gray bars representing in-distribution accuracy for Grayscale and Masked Shapes are much lower (78.8% and 53.5% respectively). Despite this high in-distribution accuracy on Color Shapes, performance drops to 66.2% and 62.6% when generalizing out-of-distribution to Grayscale and Masked Shapes, as indicated by the two lower gray bars beside the hatched green bar. This gap suggests that ViT-B/16 only learns to compare object *color* when it is trained from scratch on the Color Shapes dataset, leading to greater errors when tested on datasets that do not contain color. Still looking at Figure 7A, we can also see that fine-tuning on Masked Shapes allows for out-of-distribution generalization that is strong relative to in-distribution generalization, suggesting a more generalizable shape bias being learned by the model in this case. Figure 7B shows that CLIP pretraining weakens ViT's bias towards color, allowing for high in-distribution accuracy and near-perfect out-of-distribution generalization when trained on *any* of the three modified Shapes datasets.

For ResNet-50, we do not see evidence of an inductive bias towards color or texture when the model is trained from scratch, since bars of all colors in Figure 7C are at roughly equal heights. However, this bias slightly reappears after pretraining, with a 7.3% gap between in-distribution and Masked OOD accuracy for CLIP ResNet-50 fine-tuned on Color Shapes (Figure 7D).

## 4.2 Dissociating Color, Texture, and Shape

Results from Figure 7 suggest that some models learn to rely on certain features more than others to differentiate between objects in an image. To delve deeper into this result, we create a series of eight testing datasets based on the Shapes dataset where we vary whether shape, color, and texture are the same or different between two objects in an image (examples in Appendix B.2, Figure 24). We label each set of images with a string of letters representing whether color (C), texture (T), or shape (S) are the same. For example, images containing two objects that are the same color, different textures, and the same shape are labeled CS. CTS and "none" represent the tokens being completely the same or completely different respectively. We then evaluate the same models from Figure 7 on each of these test sets by measuring the proportion of "same" predictions for each dataset. If this proportion is high, the model views stimuli in those datasets as "same"; if this proportion is low, it means that the model views them as "different." The first few rows of Table 3 show the hypothesized behavior of theoretical models with certain inductive biases when tested on each of the generated datasets. If a model is making predictions by comparing object shape (for example), then it should predict "same" whenever the shape of the two objects in an image are the same (S) and "different" otherwise. Ideally, a model that has picked up on our definition of "same" as pixel-level similarity should not be predicting "same" for any case except for CTS.

Comparing the first row of results to predicted behavior, we immediately see that the "same" predictions made by Random ViT-B/16 on the Color Shapes dataset pattern closely with our predicted "color-biased model" behavior. This confirms our result from Figure 7, which showed that this model could not generalize to datasets without color. If the same architecture is pretrained with CLIP and *then* fine-tuned on the Color Shapes dataset, its predictions become much more sensitive to texture and shape. Upon closer inspection, however, these results reveal a bias towards color and texture that was hard to see in Figure 7. For example, CLIP ViT-B/16 classifies CT images as "same" 89% of the time when fine-tuned on Color Shapes, but only 2% of the time when it is fine-tuned on Masked Shapes. This indicates that CLIP ViT-B/16 still has an inductive bias towards color and texture during fine-tuning; it only learns to compare object shape when there are no other features available in its fine-tuning data.

In contrast, ResNet-50 has a much stronger bias towards object shape, which can be seen in Table 16 (Appendix B.2). Even when fine-tuned on Grayscale Shapes, CLIP ResNet-50 models have a preference for shape agreement. Additionally, regardless of fine-tuning dataset, ImageNet ResNet-50 models predicted "same" much less often when object shapes were mismatching. This bias towards shape is surprising considering previous results from Geirhos et al. (2019), as well as work from Raghu et al. (2021) claiming that ViT models are better at attending to global features than ResNet-50 models.

Table 3: **Predicted results of dissociation experiments compared to actual results from ViT-B/16 models fine-tuned on different versions of the original SHA dataset.** The proportion of "same" predictions for different types of images should change based on the inductive bias a given model is using. Even CLIP-pretrained ViT-B/16, which seemed from Figure 7 to be unbiased, is revealed to have a slight bias towards either color & texture or shape depending on its fine-tuning dataset. Median results over five seeds are reported for each row. Results for Random ViT-B/16 fine-tuned on Grayscale and Masked Shapes are not shown due to low accuracy (making the results difficult to interpret); full table is Table 16.

| | *Acc.* | *Proportion of "Same" Predictions* | | | | | | | |
|---|---|---|---|---|---|---|---|---|---|
| **Predicted** ↓ | acc. | none | S | T | TS | C | CS | CT | CTS |
| (no bias) | 1.00 | 0.00 | 0.00 | 0.00 | 0.00 | 0.00 | 0.00 | 0.00 | 1.00 |
| color | 1.00 | 0.00 | 0.00 | 0.00 | 0.00 | 1.00 | 1.00 | 1.00 | 1.00 |
| texture | 1.00 | 0.00 | 0.00 | 1.00 | 1.00 | 0.00 | 0.00 | 1.00 | 1.00 |
| shape | 1.00 | 0.00 | 1.00 | 0.00 | 1.00 | 0.00 | 1.00 | 0.00 | 1.00 |
| **ViT-B/16** (Rand) ↓ | acc. | none | S | T | TS | C | CS | CT | CTS |
| Color Shapes | 0.91 | 0.15 | 0.15 | 0.17 | 0.16 | 0.86 | 0.87 | 0.96 | 0.97 |
| **ViT-B/16** (CLIP) ↓ | acc. | none | S | T | TS | C | CS | CT | CTS |
| Color Shapes | 1.00 | 0.00 | 0.01 | 0.03 | 0.09 | 0.12 | 0.41 | 0.89 | 1.00 |
| Grayscale Shapes | 1.00 | 0.00 | 0.00 | 0.01 | 0.06 | 0.02 | 0.26 | 0.59 | 1.00 |
| Masked Shapes | 1.00 | 0.00 | 0.04 | 0.00 | 0.24 | 0.00 | 0.47 | 0.02 | 1.00 |

One limitation with this approach is that the difference between color and texture is somewhat ill-defined on a pixel level. This may be an explanation for the fact that no models tested exhibited a pattern close to our hypothesized "texture" model, despite evidence for "color" and "shape" being quite clear.

## 5 Related Work

**Prior work on the same-different relation.** Learning the same-different relation when the two objects being compared occupy the same image appears to be the most challenging setting for deep neural networks. Most closely related to our work is Puebla & Bowers (2022). Their setup is very similar to ours in a few respects: they define the same-different task identically to us, fine-tuning ImageNet pretrained ResNet-50 models on the task using stimuli from Fleuret et al. (2011) (which are nearly identical to our Squiggles stimuli), and testing OOD generalization on nine evaluation sets. They find that their models fail to generalize out-of-distribution and draw the conclusion that current CNNs are unable to learn the relation. Replicating their setup, we find that the difference in our results is due to the differing architectures and pretraining methods we investigated. Our ImageNet ResNet-50 model fine-tuned on Squiggles also struggles to generalize to the evaluation sets used in Puebla & Bowers (2022), but our CLIP ViT-B/16 model fine-tuned on Squiggles generalizes perfectly or nearly perfectly to seven out of nine of these sets. See Appendix A.1 for details. Additionally, Funke et al. (2021) report that ImageNet pretrained ResNet-50 models fine-tuned on stimuli from Fleuret et al. (2011) can generalize the relation to in-distribution test stimuli in this setting, but OOD generalization is not tested. The double-starred (∗∗) bar in Figure 3 replicates their results. Messina et al. (2022) show that a recurrent, hybrid CNN+ViT can attain high in-distribution test accuracy when objects occupy the same image, while (Webb et al., 2023b) demonstrate success using slot attention to segment the objects. Otherwise, successful generalization of the same-different relation with deep neural networks has been limited to a setting where objects are segmented into two separate inputs by humans and separately passed into a neural network (Kim et al., 2018; Webb et al., 2020; Kerg et al., 2022; Altabaa et al., 2023; Geiger et al., 2023).

**Abstract relation learning.** More generally, our work relates to a larger body of work concerned with the abilities of deep neural networks to learn abstract relations. The ability to abstract from sparse sensory data

is theorized to be fundamental to human intelligence (Tenenbaum et al., 2011; Ho, 2019) and is strengthened by the acquisition of language (Gentner & Hoyos, 2017). In contrast, standard deep neural networks often struggle to learn relational reasoning from sensory data alone, even when the training corpus is very large (Mitchell, 2021; Davidson et al., 2023). This is often pointed to as a key discrepancy between human and machine visual systems.

## 6 Discussion and Conclusion

Previous work has argued that deep neural networks struggle to learn the same-different relation between two objects in the same image (Kim et al., 2018; Puebla & Bowers, 2022), but the scope and nature of these difficulties are not fully understood. In this article, we tested several architectures with a number of pretraining methods and fine-tuning datasets in order to investigate the ability of neural networks to learn and generalize the same-different relation. Some of our model configurations are able to generalize the relation across all of our out-of-distribution evaluation datasets; the best model is CLIP-pretrained ViT fine-tuned on the Squiggles same-different task. Across five random seeds, this model yields a median test accuracy of nearly 100% on every evaluation dataset we use. Furthermore, this model can generalize the relation from an artificial 2D setting to a more challenging 3D setting with up to 95% test accuracy without any additional fine-tuning on 3D stimuli. The existence of such a model suggests that deep neural networks can learn generalizable representations of the same-different relation, at least for the tests we examined.

There are a number of possible reasons why CLIP-pretrained Vision Transformers exhibit the strongest out-of-distribution generalization. CLIP pretraining may be helpful because of the diversity of the dataset, which Fang et al. (2022) argue is key in the robust generalization of CLIP models in other settings. Another hypothesis is that linguistic supervision from captions containing phrases like "same," "different," or "two of" (which ImageNet-supervised or self-supervised models would have no exposure to) helps models to separate same and different objects in their visual embedding spaces, an idea supported by the results of our linear probe experiments (Appendix A.7). Separately, ViTs may perform the best on the same-different task because of their larger receptive field size; CNNs can only compare distant image patches in deeper layers, whereas ViTs can compare any image patch to any other as early as the first self-attention layer. Thus, ViTs may be able to integrate complex shape information and compare individual objects to each other more efficiently than CNNs. A combination of these pretraining and architecture benefits may explain CLIP ViT's advantage.

What mechanisms enable certain models to generalize the same-different relation? It is possible that models learn an internal circuit (e.g. Nanda et al. (2023)) in which they segment two objects and compute their equality in embedding space, implicitly implementing the components of relational architectures from earlier works that explicitly separate objects. Equality could plausibly be computed in two different ways: 1) a "perfect" comparison, where exact equality between all pixels is required, or 2) a "fuzzy" comparison in embedding space, where objects exceeding a similarity threshold are considered "same." Since models are fine-tuned on pixel-level same-different tasks, one might expect successful models to learn a "perfect" comparison. However, evidence points towards CLIP ViT learning a "fuzzy" notion of equality. First, a perfect pixel-level comparison would fail on the photorealistic stimuli, while a "fuzzy" comparison would succeed with an appropriate threshold. Secondly, models reliably classify OOD "different" objects as "same" if the objects are more similar to each other in embedding space compared to the objects the model is fine-tuned on; we observe this pattern using 13 additional same-different evaluation sets from Puebla & Bowers (2022) (see Appendix A.1). CLIP ViT-B/16 fine-tuned on SQU fails to generalize to datasets where objects have significantly higher cosine similarity in embedding space compared to SQU, but this appears to be caused by a similarity threshold that is set too low rather than a failure to generalize same-different.

As our results strongly suggest that neural networks *can* learn generalizable same-different relations, the next step for future work is to investigate the internal workings of successful models. We believe that further progress in understanding abstraction ought to come from circuit-level investigations into the same-different relation, as well as other abstract visual relations. Such investigations may reveal additional relational reasoning capabilities that have long been considered out of reach for standard deep neural networks.

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

# A    Additional Generalization Results

## A.1    Testing Models on Evaluation Sets from Puebla & Bowers (2022)

We test two of our models on the evaluation sets from Puebla & Bowers (2022): ImageNet ResNet-50 fine-tuned on SQU, which is roughly equivalent to the models tested in Puebla & Bowers (2022), and CLIP ViT-B/16 fine-tuned on SQU, which is our best model. We use code from Puebla & Bowers (2022) to generate test sets of 6,400 images evenly split between the classes, which is equal to the size of our test sets. Figure 8 show all 10 evaluation datasets used in this section. Furthermore, we report median AUC-ROC to better match Puebla & Bowers (2022), who report mean AUC-ROC. The rest of our methodology follows Section 2. We also test models on four more challenging evaluation sets from Puebla & Bowers (2022); details and results can be found in Subsection A.1.1.

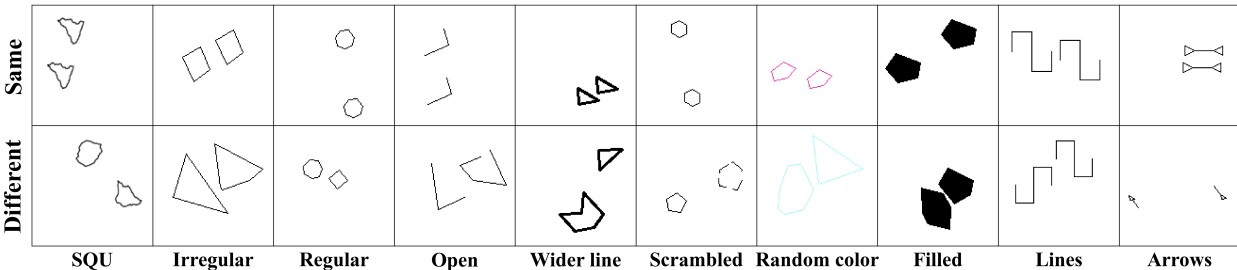

Figure 8: **Examples of "same" and "different" stimuli from all 10 evaluation sets in Figure 9.**
The first dataset (SQU) is the in-distribution test set and is the same as our SQU dataset from the main
body of this paper. The other nine datasets are generated following Puebla & Bowers (2022).

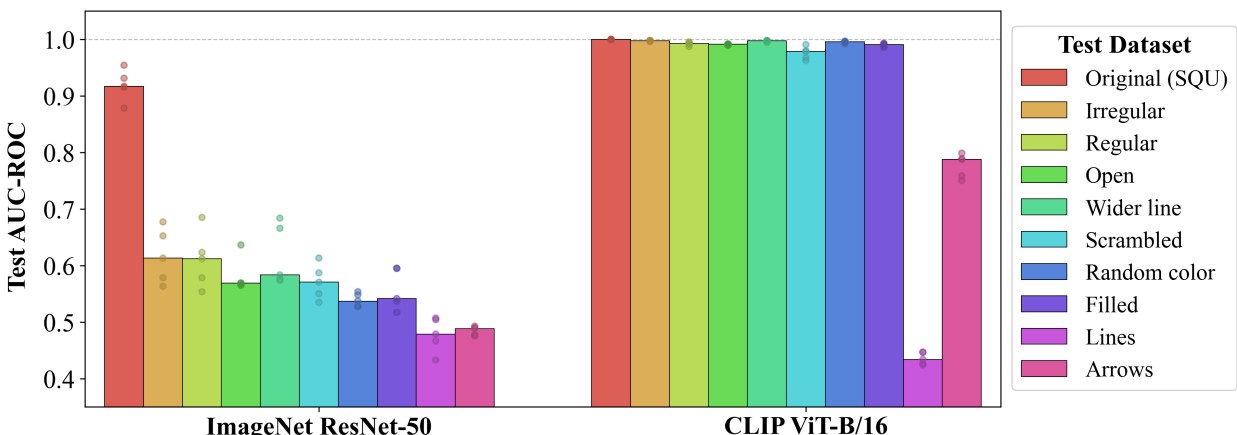

Figure 9: **Out-of-distribution test AUC-ROC for ImageNet ResNet-50 and CLIP ViT-B/16
fine-tuned on Squiggles.** Median AUC-ROC over five seeds is reported with individual runs also shown.
The legend on the right indicates the test dataset. The two red bars in this figure show in-distribution test
AUC-ROC, which is also reported for CLIP ViT-B/16 in Table 12.

We test two of our models on the 9 main evaluation sets from Puebla & Bowers (2022): ImageNet ResNet-
50 fine-tuned on SQU, which is roughly equivalent to the models tested in Puebla & Bowers (2022), and
CLIP ViT-B/16 fine-tuned on SQU, which is our best model. We use code from Puebla & Bowers (2022) to
generate test sets of 6,400 images evenly split between the classes, which is equal to the size of our test sets.
Figure 8 show all 10 evaluation datasets used in this section. Furthermore, we report median AUC-ROC to
better match Puebla & Bowers (2022), who report mean AUC-ROC. The rest of our methodology follows
Section 2.

Our ImageNet ResNet-50 results are comparable to results from Puebla & Bowers (2022) but not identical.
Differences in our specific results may be due to our differing methods for creating our datasets. For example,
the sizes of their objects are variable and may either be smaller or larger than our chosen size of $64x64$ pixels
(see Figure 8 for examples). Their ImageNet ResNet-50 model is fine-tuned on Fleuret et al. (2011) stimuli
in which the sizes of the objects also vary, whereas our model is fine-tuned on objects of a fixed size. We
also thicken the lines of our SQU stimuli, while Puebla & Bowers (2022) do not. Furthermore, they use
more fine-tuning images than us (28,000 versus 6,400), and their hyperparameters likely differ as well. Even
still, the larger pattern of results is the same—ImageNet ResNet-50 fine-tuned on the same-different relation
using stimuli from Fleuret et al. (2011) (our SQU stimuli) attains relatively high in-distribution test accuracy
but struggles to generalize out-of-distribution. This agrees with the results we obtain using our evaluation
sets (SQU, ALPH, SHA, & NAT); Table 7 shows that ImageNet-ResNet-50 fine-tuned on SQU struggles to
generalize out-of-distribution.

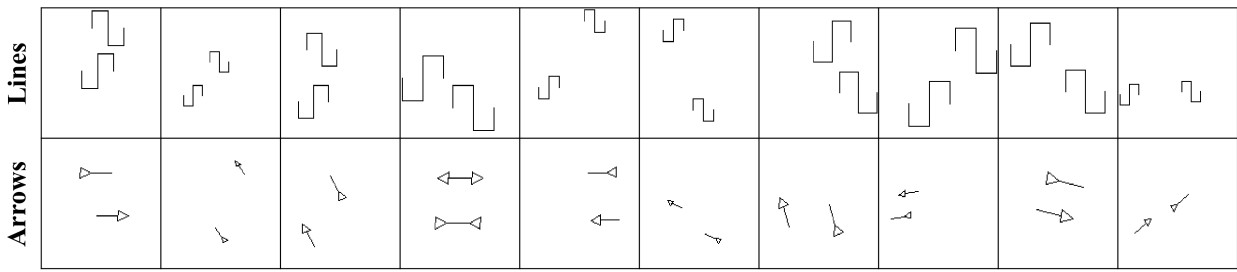

Figure 10: **"Different" images misclassified by CLIP ViT-B/16 as "same" from Puebla & Bowers (2022)'s Lines and Arrows datasets.** These stimuli are randomly sampled from the set of stimuli misclassified by all five seeds. Nearly 100% of model errors across evaluation datasets and seeds are mistaking "different" stimuli for "same" stimuli, so we only show mistakes of this kind. Note that the "different" Lines stimuli (middle row) are actually the same under reflection. Confusion matrices computed on these two datasets for the models tested in this section can be found in Appendix A.6.

In contrast, CLIP ViT-B/16 fine-tuned on our Squiggles dataset achieves perfect or nearly perfect in- and out-of-distribution generalization, with the exception of two test datasets (Lines and Arrows). This performance is rather remarkable given that objects in the evaluation datasets from Puebla & Bowers (2022) vary greatly in size, whereas our CLIP ViT-B/16 model is fine-tuned on objects of a fixed size only. This suggests that CLIP ViT-B/16 fine-tuned on SQU may learn a same-different relation that is invariant to certain qualities (such as object size) without explicit fine-tuning for such invariance. This is also supported by CLIP ViT's generalization to photorealistic stimuli in Section 3.3, in which objects vary in size and pose. Figure 10 shows examples of stimuli from the two more challenging datasets (Lines and Arrows) for which all five CLIP ViT-B/16 random seeds make errors. For Arrows, this lack of generalization may be due to symbols overlapping or being much closer to each other than any stimuli in our fine-tuning data. It's also possible that the model lacks the spatial reasoning required to form useful object representations for the Arrows dataset—unlike other datasets, this dataset requires the ability to reason about the direction of an identical line relative to identical triangles in order to distinguish objects. Thus, failure on Arrows may simply be due to "perceptual" errors like difficulties in segmenting the objects or failures in spatial reasoning rather than a lack of a general same-different representation. We also see a very slight decrease in test AUC-ROC for the Scrambled dataset, which is an interesting case. Errors made for this dataset were primarily due to our model misclassifying slightly scrambled and unmodified polygons as the "same." This error may offer insight into how exactly CLIP ViT-B/16 fine-tuned on Squiggles compares objects in an image.

However, the most surprising finding is that CLIP ViT-B/16 classifies all stimuli in the Lines dataset as "same" (see Appendix A.6), and that its ROC-AUC score is below 0.5. This is striking because all objects in the Lines dataset *are* actually the same under reflection. This result makes it very tempting to conclude that CLIP ViT-B/16 actually learns to generalize to reflections without ever being fine-tuned to do so. In fact, if models see the same image multiple times but flipped horizontally during pretraining—which is a common data augmentation—then pretrained models may already have reflection invariance baked in. Pretraining data augmentations have been shown to have such an effect on other abstract relational learning tasks (Davidson et al., 2023). While a proper treatment of CLIP's invariances are outside of the scope of this work, we test our intuition that CLIP ViTs learn a reflection-invariant same-different relation in Subsection A.1.3 below. We find evidence that strongly suggests our intuition. However, there are other possible contributing factors to CLIP ViT's failure on the Lines dataset. For instance, the Lines dataset consists entirely of one unique object that is scaled and flipped to create all stimuli; therefore, if our model makes a perceptual error when processing this particular object, that error could plausibly occur across the entire dataset.

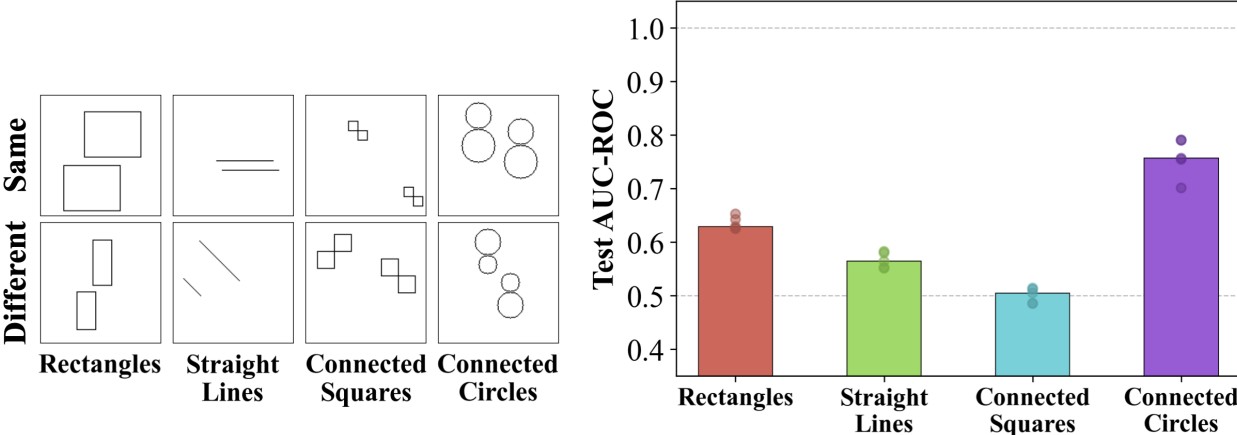

Figure 11: **Example stimuli (left) and median test AUC-ROC scores for CLIP ViT-B/16 fine-tuned on SQU (right) on four more challenging evaluation sets from Puebla & Bowers (2022).** Left: "same" stimuli are displayed in the top row, while "different" stimuli are in the bottom row. Note that "different" stimuli in the Connected Squares and Connected Circles datasets are actually the same under reflection. Right: CLIP ViT's generalization to these four datasets is notably worse than the other datasets tested in Appendix A.1. There are a number of possible explanations; see Subsections A.1.2 and A.1.3.

Table 4: **Model classifications and average logits by ground-truth class for CLIP ViT-B/16 fine-tuned on SQU on the four datasets in Figure 11.** The % Pred. "Same" column indicates the percentage of all stimuli (which are evenly split between "same" and "different") are predicted "same." Nearly 100% of stimuli for each dataset receive a "same" classification. The GT "Same" and "Diff" Logit columns indicate the model's average "same" logit for ground truth "same" and "different" images respectively. Images that are actually the same receive reliably stronger "same" judgements except in the case of Connected Squares.

| Dataset ↓ | % Pred. "Same" | GT "Same" Logit | GT "Diff" Logit |
|---|---|---|---|
| Rectangles | 99.97 | 3.82 | 3.45 |
| Straight Lines | 99.98 | 3.81 | 3.61 |
| Connected Squares | 100.0 | 3.73 | 3.76 |
| Connected Circles | 99.95 | 4.0 | 3.2 |

### A.1.1 More Challenging Datasets from Puebla & Bowers (2022)

In order to better understand CLIP ViT's limitations, we further test the ability of CLIP ViT-B/16 fine-tuned on SQU (our best model from previous sections) to generalize to four additional datasets from Puebla & Bowers (2022): Rectangles, Straight Lines, Connected Squares, and Connected Circles (see Figure 11 for example stimuli). These datasets are somewhat more challenging than previously tested datasets because they either contain extremely minimal visual information (Rectangles and Straight Lines) or require models to correctly process objects consisting of two sub-objects (Connected Squares and Connected Circles). Our methodology is exactly the same as described earlier in Appendix A.1.

Results for the same five model seeds in Figure 9 (and the main body of the paper) are presented in the bar chart in Figure 11. Performance is slightly above chance for the Rectangles and Straight Lines datasets, exactly chance for Connected Squares, and well above chance for Connected Circles (although not near perfect or excellent). At first, these results appear to contradict our main claim: that CLIP ViT-B/16 fine-tuned on SQU learns a generalizable representation of same-different. However, we believe that the failure of CLIP ViT to generalize to these datasets is actually a result of the model's "fuzzy" same-different computation rather than an abject failure to generalize the relation. Instead of computing a perfect equality between each pixel of the two objects, CLIP ViT appears to use an embedding similarity threshold to determine sameness. This can lead to model errors when the learned threshold is too low for a new OOD dataset.

Our first line of evidence that this is the case is CLIP ViT's strong performance on the photorealistic dataset in Section 3.3. The "same" objects in these images are not the same on a pixel level, yet CLIP ViT can still accurately classify them. This strong performance could only be enabled by a "fuzzy" same-different computation whereby exact pixel-level details are disregarded. Note that the objects in the photorealistic dataset can vary greatly in size and pose; the objects in the Rectangles and Straight Lines datasets (Figure 11) are more or less the same entity but varied in size (and in the case of Rectangles, "pose," due to the slightly different height-width ratios of the rectangles). Thus, this size and pose invariance could explain the model's poor generalization to Rectangles and Straight Lines.

Our second line of evidence is the distribution of CLIP ViT logits on the Rectangles, Straight Lines, and Connected Circles datasets. We examine the logits of the median seed of CLIP ViT fine-tuned on SQU (i.e. the seed corresponding to the bars in Figure 11) on these four datasets. See Table 4 for results. First, we note that the model predicts "same" for nearly 100% of the images in each dataset (% Pred. "Same" in Table 4). However, the strength of these "same" classifications differs reliably between ground truth "same" and "different" images for three of the four datasets. The average "same" logit for truly "same" images (GT "Same" Logit in Table 4) in the Rectangles, Straight Lines, and Connected Circles datasets is higher than the average "same" logit for "different" images (GT "Diff" Logit in Table 4). This indicates that the model does in fact discriminate between "same" and "different" to some extent for these datasets.

Our third and final line of evidence is the relationship of the average cosine similarity between "different" object embeddings in a given dataset and CLIP ViT SQU's performance on the dataset. Objects that are considered on average much more similar according to CLIP ViT SQU than the objects in the SQU dataset predict poor generalization performance; this is likely because these objects exceed the model's learned threshold for judging "same," which is calibrated for SQU stimuli. See Subsection A.1.2.

Separately, note that CLIP ViT appears to learn a same-different relation that is invariant to reflection; in other words, the same object reflected is still considered "same" when compared to the non-reflected version. The "different" images in the Lines, Connected Squares, and Connected Circles datasets are in fact the same object reflected. In fact, the two most difficult datasets for CLIP ViT—Lines and Connected Squares—both feature the same type of reflection: reflection across the $y$-axis of the objects. We test CLIP ViT SQU on a version of our fine-tuning datasets where objects are reflected across the $y$-axis, finding that this drops model generalization to these datasets from near perfect to near chance. This strongly suggests reflection-invariance in this model. See Subsection A.1.3 below.

### A.1.2 Object Embedding Similarity Predicts Generalization for CLIP ViT SQU

We seek to measure how visually distinct the OOD objects in the Puebla & Bowers (2022) datasets are according to a CLIP ViT fine-tuned on SQU compared to the objects in the model's fine-tuning dataset (SQU). We hypothesize that because of the model's "fuzzy" same-different computation, it will perform worse on datasets that contain objects that are more visually similar to each other compared to SQU objects.

We measure inter-object similarity by creating separate input images for each individual object. These images are equivalent to the "same" and "different" stimuli, except only one object is present. The single object is randomly placed in the image. In the case of the datasets from Puebla & Bowers (2022), we generate $1,000$ unique object images in this way for each dataset. In the case of our SQU fine-tuning set, we source $1,000$ unique objects not seen during fine-tuning. For each dataset, we embed each single-object image using the image encoder from CLIP ViT fine-tuned on SQU. We then compute pairwise cosine similarity between the embeddings of all different objects. Finally, we report the average of the pairwise object cosine similarity scores as the solid green line in Figure 12.

The green star in Figure 12 marks the average inter-object embedding similarity for unseen SQU objects, while the dashed green line indicates the maximal average inter-object embedding similarity for which CLIP ViT SQU performs well (this corresponds to the Wider dataset; see Figure 8). The datasets with an average inter-object embedding similarity above this threshold are Lines, Connected Squares (C. Squares), Straight Lines (S. Lines), Rectangles, and Connected Circles (C. Circles). Plotted in magenta in Figure 12 are the median test AUC-ROC scores on each evaluation dataset. Model performance is near perfect for all datasets

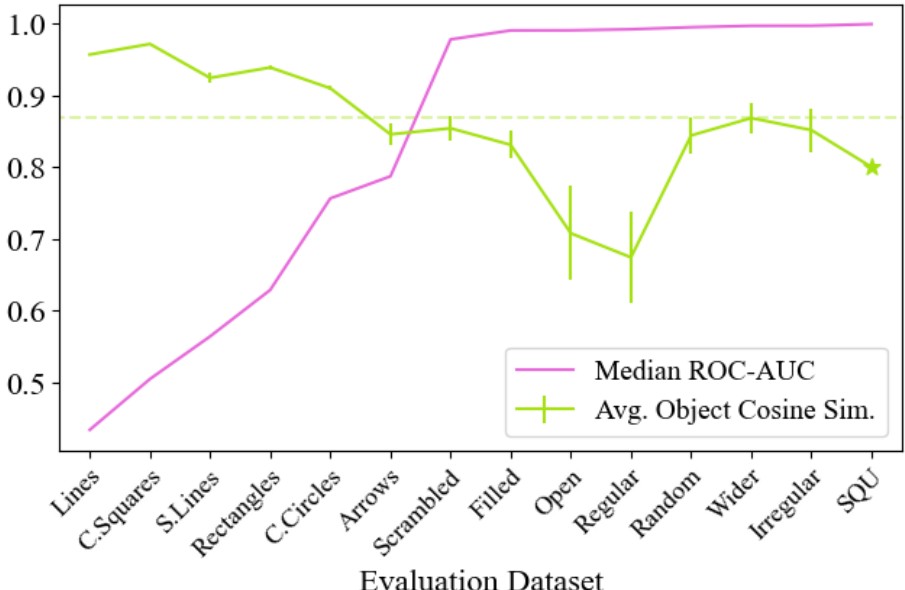

Figure 12: **Average inter-object CLIP embedding similarity for each evaluation dataset (green) vs. CLIP ViT generalization performance (magenta).** The vertical green lines represent the variance in inter-object similarity for each dataset. If different objects are overly similar to each other compared to the fine-tuning data (SQU; green star), model generalization performance drops significantly due to misclassifying all "different" images as "same."

with object similarities below the threshold marked by the dashed green line (with the exception of Arrows). Model performance drops precipitously for datasets with object similarities above the threshold. Essentially, the objects in these datasets are significantly more similar to each other compared to the objects in the fine-tuning data (SQU); thus, because CLIP ViT learns a "fuzzy" same-different computation, it considers the objects in these high-similarity datasets to be the "same" according to the lower threshold learned on SQU. Model predictions for these high-similarity datasets accord with this interpretation; nearly all "different" images are misclassified as "same" (see Table 4 as well as Appendix A.6).

### A.1.3  Reflection Invariance in CLIP ViT

The poor generalization performance of CLIP ViT on the Lines, Connected Squares, and Connected Circles datasets from Puebla & Bowers (2022) (see Figures 8 and 11) suggest an interesting possibility: that CLIP ViT learns a reflection invariant same-different relation, despite not being trained to do so (see the end of Appendix A.1). To test this, we evaluate CLIP ViTs fine-tuned on SQU on "flipped" versions of our SQU, ALPH, and NAT fine-tuning datasets. We skip the SHA dataset since many of the shapes have bilateral symmetry. The "same" stimuli in the flipped datasets are the same as the regular datasets; the "different" stimuli however are created by reflecting a copy of a given object about its $y$-axis. See the right side of Figure 13 for example stimuli. Note that this matches the definition of "different" used

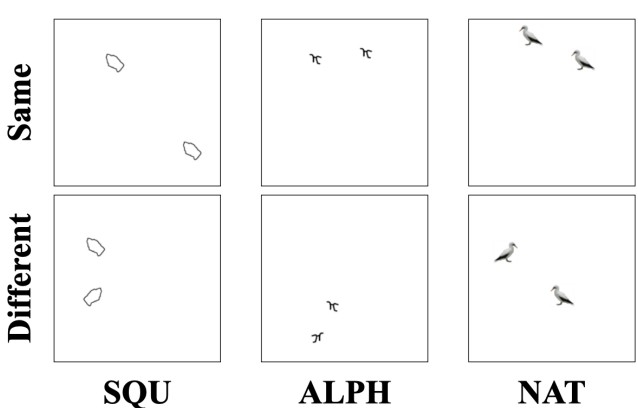

Figure 13: **Example stimuli from each "flipped" dataset.**

by the Lines, Connected Squares, and Connected
Circles datasets from Puebla & Bowers (2022).

We create flipped SQU, ALPH, and NAT datasets containing 6,400 images each, evenly divided between "same" and "different." We then compute test accuracy on these datasets for CLIP ViT-B/16 fine-tuned on SQU using the same five seeds used elsewhere in the paper. We find that median model performance drops significantly for the flipped datasets due to models predicting "same" for "different" images, indicating that the model considers reflected versions of the same object to be the same. This effect is more severe for the two OOD datasets (ALPH and NAT). Decreases in median test accuracy as well as the percentage of all stimuli predicted "same" for each dataset are the following: 99.6% (original) to 77.1% (flipped) test accuracy for SQU, with 72.9% predicted "same;" 97.7% (original) to 51.6% (flipped) test accuracy for ALPH, with 98.4% predicted "same;" 96.7% (original) to 51.8% (flipped) for NAT, with 98.2% predicted "same." This invariance likely helps to explain why model generalization suffers on Lines, Connected Squares, and Connected Circles from Puebla & Bowers (2022).

## A.2 In-Distribution Learning Curves

For each architecture and pretraining method, we plot loss and in-distribution validation accuracy per epoch of fine-tuning or training on each dataset. Lines show averages for the same set of hyperparameters (for that model & dataset) across five seeds.

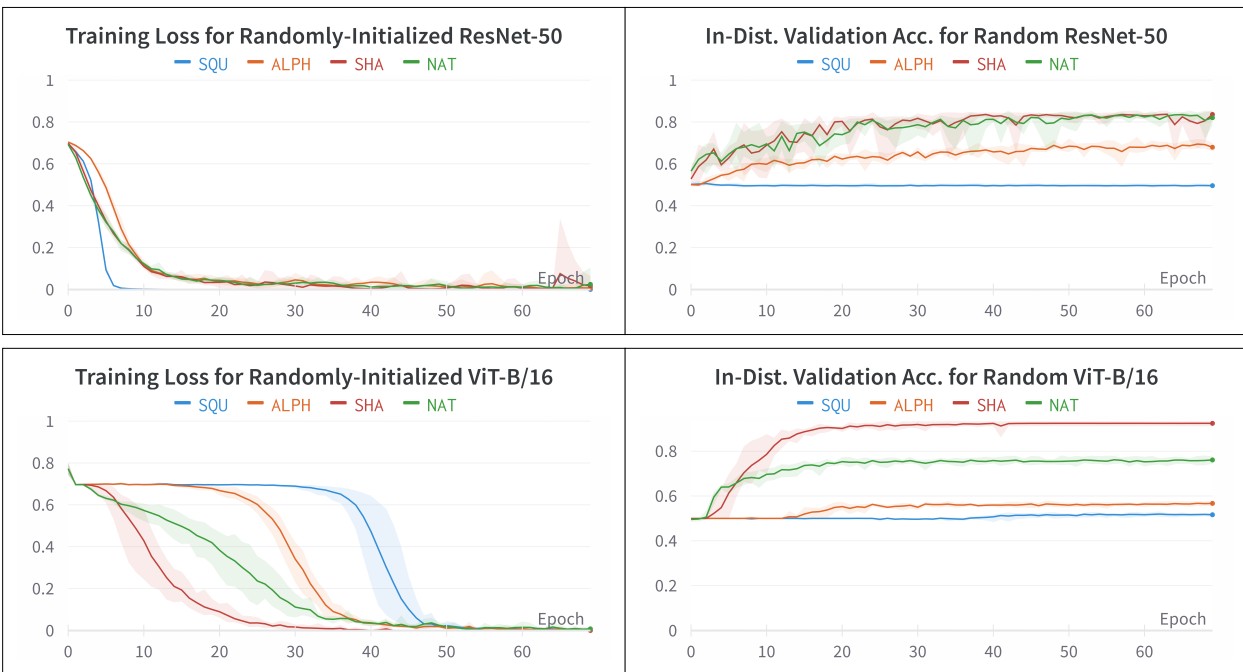

Figure 14: **Average loss curves for randomly-initialized ResNet-50 and ViT-B/16 trained on each dataset.** Even though loss curves for models trained on SQU go to zero, validation accuracy remains flat, indicating that models memorize training data. Furthermore, the loss curves for randomly-initialized ViT-B/16 distinctly mirror the hierarchy of dataset difficulty discussed in Section 3.2.

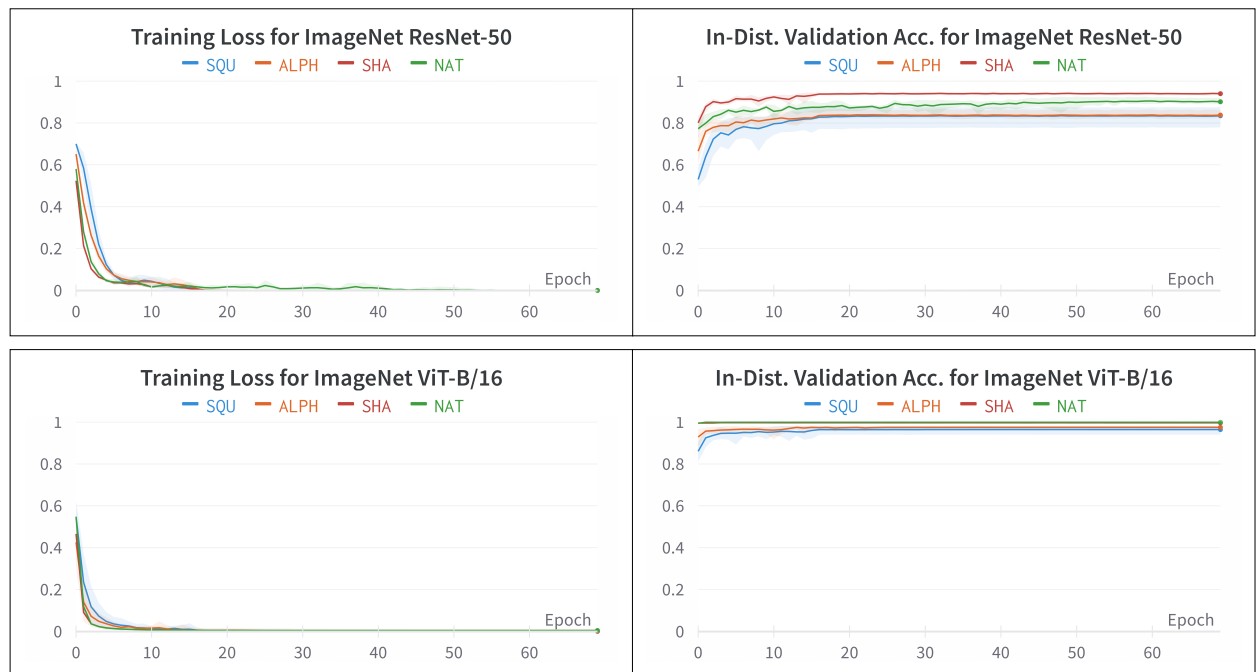

Figure 15: **Average loss curves for ImageNet-pretrained ResNet-50 and ViT-B/16 fine-tuned on each dataset.** Models converge substantially faster than in Figure 14. ImageNet ViT-B/16 models fine-tuned on SHA and NAT already attain nearly 100% validation accuracy after only one epoch.

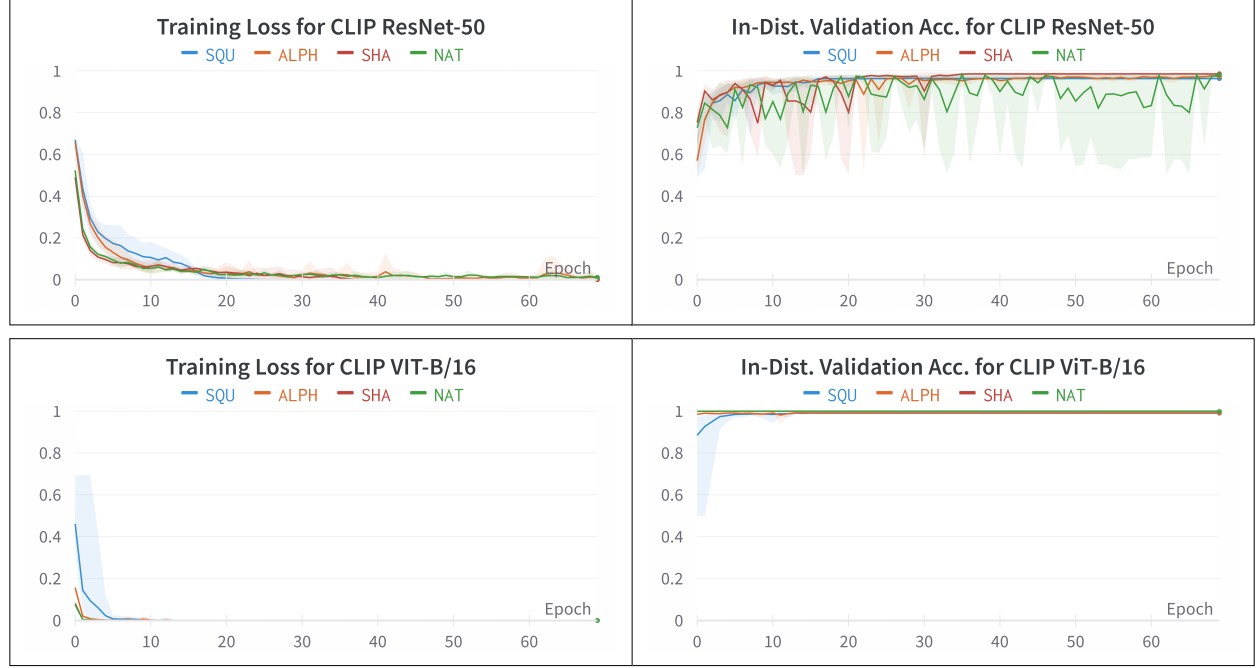

Figure 16: **Average loss curves for CLIP-pretrained ResNet-50 and ViT-B/16 fine-tuned on each dataset.** In-distribution generalization to color-containing datasets SHA and NAT seem much more difficult for CLIP ResNet-50 than CLIP ViT-B/16 (or any other model configuration). CLIP ViT-B/16 attains nearly 100% validation accuracy after only one epoch of fine-tuning on all datasets except SQU.

### A.3 Out-of-distribution Generalization Tables

We report median test accuracy over five random seeds for each pretraining method, architecture, and fine-tuning dataset. The tables below include the four main fine-tuning datasets (SQU, ALPH, SHA, NAT; see Figure 2), the grayscale and masked versions of the SHA dataset (SHA-G and SHA-M; see Figure 6a), and grayscale and masked versions of the NAT dataset (NAT-G and NAT-M). As in Table 1, rows indicate the dataset that models are fine-tuned on, while columns indicate the test dataset. The rightmost column labeled "Avg." is the row-wise average of accuracy scores across OOD evaluation sets (i.e. off-diagonal values), which indicates how well a model fine-tuned on a given dataset is able to generalize to other datasets. The bottom row labeled "Avg." is the column-wise average across off-diagonal values, indicating how difficult it is for models fine-tuned on other datasets to generalize to the given dataset.

Table 5: **OOD test accuracy for CLIP ResNet-50 models fine-tuned on each dataset.** The model fine-tuned on NAT-G exhibits the strongest average OOD generalization, although it fails to generalize to the SQU stimuli.

| | | | | CLIP ResNet-50 | | | | | |
|---|---|---|---|---|---|---|---|---|---|
| | | | | $\leftarrow$ **Test** $\rightarrow$ | | | | | |
| **Train** ↓ | SQU | ALPH | SHA | SHA-G | SHA-M | NAT | NAT-G | NAT-M | Avg. |
| SQU | **97.7** | 80.8 | 82.9 | 81.9 | 73.6 | 82.0 | 86.6 | 82.6 | 81.5 |
| ALPH | 83.5 | **97.4** | 88.9 | 90.1 | 92.9 | 90.7 | 78.2 | 83.8 | 86.9 |
| SHA | 51.3 | 69.3 | **98.1** | 96.2 | 90.8 | 95.2 | 76.3 | 86.6 | 80.8 |
| SHA-G | 65.5 | 80.7 | 98.1 | **98.2** | 95.1 | 93.7 | 95.8 | 91.5 | 88.6 |
| SHA-M | 55.9 | 68.1 | 94.7 | 92.1 | **76.1** | 79.6 | 100 | 86.4 | 82.4 |
| NAT | 53.4 | 76.0 | 95.2 | 96.1 | 96.1 | **97.3** | 87.0 | 94.3 | 85.4 |
| NAT-G | 55.6 | 81.3 | 95.4 | 97.3 | 98.0 | 95.7 | **89.7** | 92.7 | 88.0 |
| NAT-M | 59.8 | 80.6 | 90.6 | 91.4 | 90.1 | 94.8 | 94.3 | **95.0** | 85.9 |
| Avg. | 60.7 | 76.7 | 92.3 | 92.2 | 90.9 | 90.2 | 88.3 | 88.2 | |

Table 6: **OOD test accuracy for CLIP ViT-B/16 models fine-tuned on each dataset.** It is interesting to note the different patterns of generalization between models fine-tuned on SHA, SHA-G, and SHA-M. Models fine-tuned on the SHA dataset (which contains color and texture) do not generalize very well to NAT-G and NAT-M datasets; models fine-tuned on SHA-G (which removes color) generalize somewhat better to NAT-G and NAT-M; and models fine-tuned on SHA-M (which removes color and texture) attain 100% or near 100% accuracy on NAT-G and NAT-M. The same pattern holds for models fine-tuned on NAT, NAT-G, and NAT-M tasks.

| | | | | CLIP ViT-B/16 | | | | | |
|---|---|---|---|---|---|---|---|---|---|
| | | | | $\leftarrow$ **Test** $\rightarrow$ | | | | | |
| **Train** ↓ | SQU | ALPH | SHA | SHA-G | SHA-M | NAT | NAT-G | NAT-M | Avg. |
| SQU | **99.5** | 97.7 | 99.1 | 98.9 | 94.8 | 95.5 | 95.0 | 98.1 | 97.0 |
| ALPH | 59.5 | **99.4** | 99.9 | 99.9 | 98.8 | 99.7 | 95.1 | 97.5 | 92.9 |
| SHA | 50.0 | 56.0 | **100** | 98.6 | 98.2 | 100 | 60.6 | 77.7 | 77.3 |
| SHA-G | 50.2 | 63.5 | 100 | **99.9** | 99.9 | 100 | 85.5 | 95.6 | 85.0 |
| SHA-M | 55.6 | 93.3 | 100 | 100 | **99.8** | 100 | 100 | 97.8 | 92.4 |
| NAT | 50.0 | 68.4 | 99.8 | 97.8 | 99.3 | **100** | 63.0 | 83.7 | 80.3 |
| NAT-G | 50.2 | 70.6 | 99.9 | 98.9 | 100 | 100 | **71.5** | 93.9 | 87.6 |
| NAT-M | 60.2 | 92.7 | 100 | 99.9 | 100 | 100 | 94.3 | **98.5** | 92.4 |
| Avg. | 53.7 | 77.5 | 99.8 | 99.1 | 98.7 | 99.3 | 84.8 | 92.0 | |

Table 7: **OOD test accuracy for ImageNet ResNet-50 models fine-tuned on each dataset.** Unlike CLIP-pretrained models, ImageNet ResNet-50 fine-tuned on SQU actually exhibits the weakest OOD generalization.

| | | | | | ImageNet ResNet-50 | | | | |
| --- | --- | --- | --- | --- | --- | --- | --- | --- | --- |
| | | | | | ← Test → | | | | |
| Train ↓ | SQU | ALPH | SHA | SHA-G | SHA-M | NAT | NAT-G | NAT-M | Avg. |
| SQU | **84.8** | 57.4 | 59.3 | 52.6 | 65.1 | 62.9 | 50.2 | 60.8 | 58.3 |
| ALPH | 61.3 | **83.7** | 60.4 | 69.0 | 78.5 | 70.2 | 68.0 | 73.9 | 68.8 |
| SHA | 51.2 | 66.7 | **94.4** | 90.1 | 78.4 | 84.0 | 64.1 | 66.5 | 71.6 |
| SHA-G | 53.9 | 72.6 | 70.8 | **94.6** | 84.2 | 74.2 | 90.1 | 78.9 | 75.0 |
| SHA-M | 56.2 | 68.9 | 73.7 | 92.4 | **79.4** | 68.7 | 99.3 | 79.8 | 77.0 |
| NAT | 50.3 | 58.3 | 80.4 | 69.5 | 78.6 | **90.5** | 62.4 | 70.7 | 67.2 |
| NAT-G | 50.8 | 72.2 | 70.0 | 82.8 | 89.8 | 78.2 | **69.1** | 81.1 | 75.0 |
| NAT-M | 50.1 | 74.9 | 66.9 | 76.2 | 84.0 | 74.2 | 78.9 | **88.4** | 72.2 |
| Avg. | 53.4 | 67.3 | 68.8 | 76.1 | 79.8 | 73.2 | 73.3 | 73.1 | |

Table 8: **OOD test accuracy for ImageNet ViT-B/16 models fine-tuned on each dataset.** Interestingly, models fine-tuned on SHA exhibit strong generalization to the grayscale and masked versions of that dataset (but still don't generalize to SQU or ALPH).

| | | | | | ImageNet ViT-B/16 | | | | |
| --- | --- | --- | --- | --- | --- | --- | --- | --- | --- |
| | | | | | ← Test → | | | | |
| Train ↓ | SQU | ALPH | SHA | SHA-G | SHA-M | NAT | NAT-G | NAT-M | Avg. |
| SQU | **95.4** | 65.8 | 57.6 | 53.3 | 59.7 | 60.5 | 51.8 | 66.3 | 59.3 |
| ALPH | 81.7 | **97.0** | 50.5 | 51.0 | 59.1 | 52.1 | 52.1 | 67.0 | 59.1 |
| SHA | 50.0 | 50.1 | **100** | 96.2 | 99.3 | 99.4 | 55.8 | 82.5 | 76.2 |
| SHA-G | 50.0 | 61.2 | 100 | **99.8** | 99.8 | 99.9 | 73.9 | 84.8 | 81.4 |
| SHA-M | 57.7 | 88.0 | 99.9 | 99.8 | **99.6** | 97.5 | 99.9 | 97.1 | 91.4 |
| NAT | 50.0 | 50.4 | 97.3 | 80.4 | 97.8 | **100** | 50.4 | 71.8 | 71.2 |
| NAT-G | 50.0 | 50.2 | 98.3 | 91.7 | 99.7 | 99.9 | **54.0** | 87.8 | 82.5 |
| NAT-M | 52.2 | 72.3 | 99.8 | 99.3 | 99.9 | 100 | 91.7 | **98.4** | 87.9 |
| Avg. | 55.9 | 62.6 | 86.2 | 81.7 | 87.9 | 87.1 | 68.0 | 79.6 | |

Table 9: **OOD test accuracy for randomly-initialized ResNet-50 models trained on each dataset.** Models attain surprisingly high in-distribution test accuracy for certain datasets, such as SHA and SHA-G. Models trained on SQU appear to learn nothing even though their loss curves diminish (see Figure 14). This indicates that models are memorizing training examples, which is consistent with results from prior work (e.g. Kim et al. (2018)).

| | | | | | Randomly Initialized ResNet-50 | | | | |
| --- | --- | --- | --- | --- | --- | --- | --- | --- | --- |
| | | | | | ← Test → | | | | |
| Train ↓ | SQU | ALPH | SHA | SHA-G | SHA-M | NAT | NAT-G | NAT-M | Avg. |
| SQU | **49.8** | 49.7 | 49.5 | 48.2 | 49.1 | 48.3 | 49.6 | 50.0 | 49.2 |
| ALPH | 53.1 | **69.2** | 58.9 | 59.0 | 55.2 | 58.6 | 50.0 | 50.4 | 55.0 |
| SHA | 51.2 | 69.3 | **82.6** | 80.4 | 82.3 | 82.9 | 53.8 | 61.8 | 68.8 |
| SHA-G | 50.6 | 67.2 | 85.0 | **85.5** | 87.5 | 84.0 | 59.8 | 67.4 | 71.6 |
| SHA-M | 50.0 | 57.0 | 77.3 | 77.0 | **77.0** | 75.0 | 78.3 | 74.3 | 69.9 |
| NAT | 52.9 | 69.5 | 81.6 | 80.4 | 80.3 | **80.2** | 55.4 | 68.0 | 69.7 |
| NAT-G | 51.1 | 64.2 | 77.3 | 83.6 | 82.8 | 82.5 | **61.3** | 72.5 | 73.4 |
| NAT-M | 50.0 | 59.4 | 77.2 | 79.1 | 80.3 | 79.2 | 69.2 | **74.4** | 70.6 |
| Avg. | 51.3 | 62.3 | 72.4 | 72.5 | 73.9 | 72.9 | 59.5 | 63.5 | |

Table 10: **OOD test accuracy for randomly-initialized ViT-B/16 models trained on each dataset.** Given their larger receptive field size, randomly initialized ViTs somewhat surprisingly perform worse overall than randomly initialized ResNets (Table 9).

| | | | | **Randomly Initialized ViT-B/16** | | | | | |
|---|---|---|---|---|---|---|---|---|---|
| | | | | ← **Test** → | | | | | |
| **Train ↓** | SQU | ALPH | SHA | SHA-G | SHA-M | NAT | NAT-G | NAT-M | Avg. |
| SQU | **51.7** | 51.8 | 51.0 | 53.5 | 52.7 | 53.8 | 51.4 | 53.9 | 52.6 |
| ALPH | 49.9 | **54.8** | 51.7 | 51.9 | 56.8 | 52.0 | 49.9 | 51.4 | 51.9 |
| SHA | 50.0 | 50.1 | **92.9** | 66.2 | 62.6 | 73.8 | 50.7 | 56.0 | 58.5 |
| SHA-G | 50.0 | 50.5 | 74.2 | **78.8** | 66.5 | 66.4 | 55.7 | 56.4 | 60.0 |
| SHA-M | 50.0 | 50.0 | 56.5 | 59.6 | **53.5** | 55.5 | 81.3 | 63.5 | 59.5 |
| NAT | 50.1 | 51.7 | 75.7 | 58.7 | 62.2 | **76.7** | 53.4 | 62.8 | 59.2 |
| NAT-G | 50.2 | 51.8 | 64.1 | 69.9 | 70.7 | 67.0 | **55.9** | 65.9 | 62.8 |
| NAT-M | 50.0 | 50.5 | 56.4 | 56.8 | 58.4 | 58.2 | 55.0 | **66.2** | 55.0 |
| Avg. | 50.0 | 50.9 | 61.4 | 59.5 | 61.4 | 61.0 | 56.8 | 58.6 | |

## A.4 ImageNet Models with Comparable Parameters

One explanation for the difference in performance between ResNet-50 and ViT-B/16 is the fact that ResNet-50 consists of 23M parameters, whereas ViT-B/16 has a total of 86M parameters. To explore this possibility, we fine-tune and test ConvNeXt-B Liu et al. (2022) as an example of a convolutional model with 89M parameters (comparable to ViT-B/16), as well as DeiT-S Touvron et al. (2021), a 22M parameter transformer model of similar size to ResNet-50.

Results for ConvNeXt-B are shown on the left in Table 11. This model was pre-trained on ImageNet-22k, the same dataset as ImageNet ViT-B/16, and also has a similar number of parameters as ViT-B/16. Comparing the left-hand side of Table 11 to Table 8, we can see that despite ConvNeXt's competitive edge in parameter count, ViT-B/16 still seems to have slightly stronger OOD generalization between SHA and NAT as well as between SQU and ALPH. However, ConvNeXt does generalize with a 71.7% accuracy to SHA when trained on ALPH, which is better than any other non-CLIP model tested.

Looking at the right-hand side of Table 11, we see that compared to ResNet-50 (Table 7), DeiT-S has more success generalizing within SQU and ALPH as well as SHA and NAT. These two smaller models were pre-trained on ImageNet-1k, but despite this DeiT-S is still quite competitive with the larger ConvNeXt, which was pre-trained on significantly more data (ImageNet-22k). Taken as a whole, these additional results seem to suggest that the biggest differentiator between ResNet-50 and ViT-B/16 is not parameter count or size of pretraining data but architectural design.

Table 11: **OOD test accuracy for ImageNet ConvNeXt-B and DeiT-S fine-tuned on the main four datasets.** Performance is competitive between a small transformer model of similar size to ResNet-50 and a large convolutional model comparable to ViT-B/16, suggesting that the difference mainly comes down to architecture.

| | **ImageNet-22k ConvNeXt-B** | | | | | | **ImageNet-1k DeiT-S** | | | | |
|---|---|---|---|---|---|---|---|---|---|---|---|
| | ← **Test** → | | | | | | ← **Test** → | | | | |
| **Train ↓** | SQU | ALPH | SHA | NAT | Avg. | **Train ↓** | SQU | ALPH | SHA | NAT | Avg. |
| SQU | **87.0** | 55.3 | 52.8 | 52.5 | 61.9 | SQU | **85.6** | 61.4 | 54.0 | 53.9 | 63.7 |
| ALPH | 64.0 | **93.6** | 71.7 | 61.7 | 72.7 | ALPH | 62.2 | **88.7** | 63.1 | 71.3 | 71.3 |
| SHA | 50.0 | 50.2 | **98.4** | 86.9 | 71.4 | SHA | 50.1 | 52.2 | **97.9** | 94.5 | 73.7 |
| NAT | 50.0 | 50.6 | 78.5 | **98.2** | 69.3 | NAT | 50.0 | 52.7 | 85.2 | **98.4** | 71.6 |
| Avg. | 62.8 | 62.4 | 75.4 | 74.8 | | Avg. | 62.0 | 63.7 | 75.0 | 79.5 | |

## A.5 Area Under the ROC Curve for CLIP Models

In addition to reporting median test accuracy across seeds, we report median area under the ROC curve for CLIP ResNet-50 and CLIP ViT-B/16. Table 12 below mirrors Table 1 from the main paper.

Table 12: **Out-of-distribution test AUC for CLIP models fine-tuned on each dataset.** Rows indicate the dataset that models are fine-tuned on, while columns indicate the test dataset. Each cell is the median performance over five random seeds. The rightmost column labeled "Avg." is the row-wise average of accuracy scores across OOD evaluation sets (i.e. off-diagonal values), which indicates how well a model fine-tuned on a given dataset is able to generalize to other datasets. The bottom row labeled "Avg." is the column-wise average across off-diagonal values, indicating how difficult it is for models fine-tuned on other datasets to generalize to the given dataset.

| **CLIP ResNet-50** | | | | | | **CLIP ViT-B/16** | | | | | |
| | $\leftarrow$ **Test** $\rightarrow$ | | | | | | $\leftarrow$ **Test** $\rightarrow$ | | | | |
| **Train** $\downarrow$ | SQU | ALPH | SHA | NAT | Avg. | **Train** $\downarrow$ | SQU | ALPH | SHA | NAT | Avg. |
| SQU | **0.99** | 0.95 | 0.93 | 0.86 | 0.91 | SQU | **1.00** | 1.00 | 1.00 | 1.00 | 1.00 |
| ALPH | 0.96 | **0.99** | 0.96 | 0.97 | 0.96 | ALPH | 0.93 | **1.00** | 1.00 | 1.00 | 0.98 |
| SHA | 0.8 | 0.91 | **1.0** | 0.99 | 0.9 | SHA | 0.62 | 0.91 | **1.00** | 1.00 | 0.84 |
| NAT | 0.83 | 0.94 | 0.99 | **0.99** | 0.92 | NAT | 0.63 | 0.93 | 1.00 | **1.00** | 0.85 |
| Avg. | 0.86 | 0.93 | 0.96 | 0.94 | | Avg. | 0.73 | 0.95 | 1.00 | 1.00 | |

Models fine-tuned on the Shapes and Naturalistic datasets attain rather high AUC across all OOD test datasets, notably including the Alphanumeric task (which does not contain color or texture). CLIP ResNet-50 in particular attains $> 0.8$ AUC across all fine-tuning conditions and test datasets. This is in contrast to median accuracy results reported in Table 1, which shows more of a dramatic "upper triangular" pattern. This indicates that some of the models that achieve poor OOD test accuracy may perform much more strongly with a correctly calibrated bias. Even still, the "upper triangular" pattern is still evident here—models fine-tuned on Squiggles and Alphanumeric tasks demonstrate stronger generalization than models fine-tuned on Shapes and Naturalistic tasks. Furthermore, ViT still outperforms ResNet, achieving perfect AUC across all test datasets when fine-tuned on Squiggles.

### A.6    Out-of-distribution Test Confusion Matrices

We consider the pattern of errors produced by two of our models: ImageNet ResNet-50 fine-tuned on SQU, which is the most similar to models tested in some prior work (Funke et al., 2021; Puebla & Bowers, 2022), and CLIP ViT-B/16 fine-tuned on SQU, which is our best model. We compute confusion matrices for both of these models on our four main test sets (SQU, ALPH, SHA, & NAT) as well as the Lines and Arrows test sets from Puebla & Bowers (2022), which our CLIP ViT-B/16 model finds challenging (see Appendix A.1 for visual examples and results). We report matrices for the random seed that yields the median in-distribution test accuracy (i.e. the run that corresponds to the bars in Figure 3).

In general, both ImageNet ResNet-50 and CLIP ViT-B/16 models tend to mistake "different" stimuli for "same" stimuli more frequently than the converse. However, this is not always the case for ImageNet ResNet-50—as the top row of Figure 17 shows, ResNet makes the opposite error (mistaking "same" for "different") much more frequently when tested on SHA and NAT datasets. This is never the case for CLIP ViT-B/16 (bottom row of Figure 17). Furthermore, the difference in frequency between the two types of errors is much more stark for CLIP ViT-B/16; the vast majority of errors made by this model across all test datasets are mistaking "different" stimuli for "same" stimuli. Hochmann (2021) argues that much of the studies on same-different relation learning in children and animals can actually be accounted for by subjects learning a concept of "same" without learning a symmetric concept of "different;" in other words, a subject can achieve high performance on many same-different tasks used in the cognitive science literature by only recognizing when two objects are the same as each other (without explicitly representing "different"). This seems to align with the errors made by CLIP ViT-B/16. It is possible that this model learns a stronger or more coherent concept of sameness and thus decides to output "same" whenever it is less certain.

Another notable result is the CLIP ViT-B/16 confusion matrix for the Lines dataset from Puebla & Bowers (2022). The model assigns the label "same" to 100% of the "different" stimuli with relatively high confidence (as indicated by the $< 0.5$ AUC-ROC score on this dataset in Appendix A.1). This is in contrast to ImageNet ResNet-50, which appears to assign category labels at random for the Lines dataset. As extrapolated in

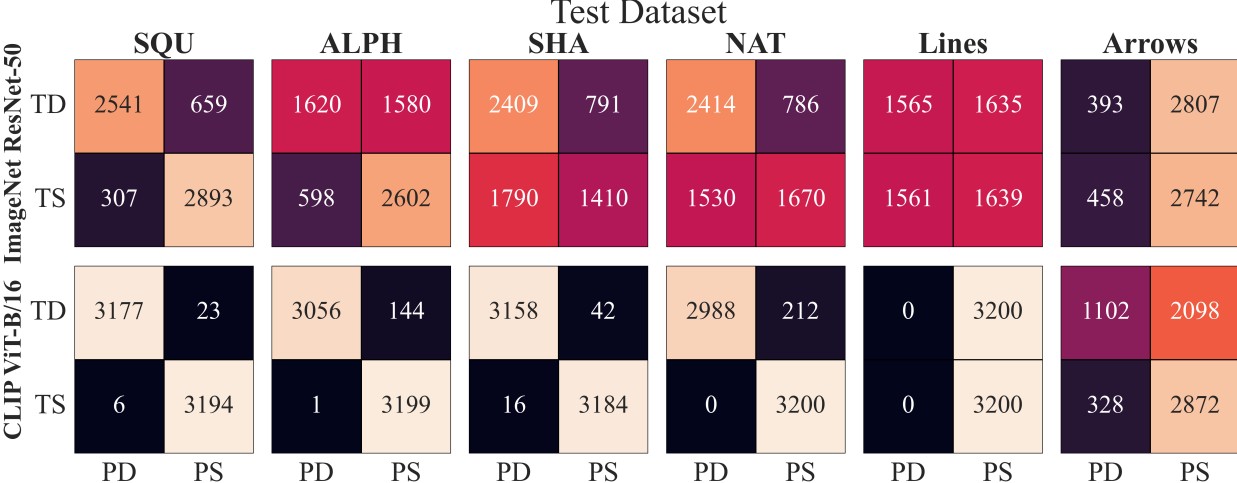

Figure 17: **Confusion matrices for ImageNet ResNet-50 (top row) and CLIP ViT-B/16 (bottom row) fine-tuned on SQU.** Each column gives confusion matrices for a given test set as indicated by the labels above. The rows of the confusion matrices are the true labels (TD means "true different"; TS means "true same"), while the columns of the matrices are the predicted classes (PD means "predicted different"; PS means "predicted same"). Each cell in the matrix shows the number of test images with a given true label and a predicted label as assigned by each model.

Appendix A.1, the "different" stimuli in this dataset are actually the same under reflection, suggesting that CLIP ViT-B/16 fine-tuned on SQU may learn a reflection-invariant same-different relation despite not being fine-tuned for such invariance (although this is speculative).

## A.7 Probing CLIP Embeddings

In order to determine the degree to which CLIP pretraining alone encodes useful information for learning the same-different relation, we perform a linear probe on the CLIP ResNet-50 and CLIP ViT-B/16 models. As in our main experiments, we append a linear binary classifier to the visual backbone of each model. However, in this experiment, we freeze the pretrained weights in the backbone of each model and train only the parameters of the classifier on the fixed embeddings given by the backbone. Results are displayed in Table 13.

Table 13: **Out-of-distribution test accuracy for the best linear probe trained on CLIP embeddings of each dataset.**

| | CLIP ResNet-50 Probe | | | | | | CLIP ViT-B/16 Probe | | | | |
|---|---|---|---|---|---|---|---|---|---|---|---|
| | | ← Test → | | | | | | ← Test → | | | |
| Train ↓ | SQU | ALPH | SHA | NAT | Avg. | Train ↓ | SQU | ALPH | SHA | NAT | Avg. |
| SQU | **62.4** | 50.0 | 50.0 | 50.0 | 50.0 | SQU | **81.9** | 51.1 | 55.8 | 52.7 | 53.2 |
| ALPH | 50.0 | **72.7** | 50.1 | 49.8 | 49.9 | ALPH | 50.0 | **94.4** | 53.1 | 58.5 | 53.9 |
| SHA | 50.0 | 50.0 | **85.6** | 50.3 | 50.1 | SHA | 50.0 | 50.0 | **99.9** | 90.4 | 63.5 |
| NAT | 50.0 | 49.9 | 52.5 | **85.6** | 50.8 | NAT | 50.0 | 50.1 | 70.6 | **100** | 56.9 |
| Avg. | 50.0 | 50.0 | 50.8 | 50.0 | | Avg. | 50.0 | 50.4 | 59.8 | 67.2 | |

We find that the linear probe can generally exhibit rather high in-distribution generalization. CLIP embeddings of Naturalistic stimuli produce the highest in-distribution test accuracy, followed closely by Shapes. CLIP embeddings of Alphanumeric and Squiggles datasets are more difficult to learn from. This mirrors the ordering observed in Section 3.2 in which the two same-different tasks containing color and texture features tend to be easier to learn, while the shape-based tasks tend to be more difficult. The fact that Alphanumeric

and Squiggles probes are unable to generalize OOD, however, is odd considering the fact that the solutions to both of these datasets should be the same (based on shape); this implies there is some other signal that linear probes are picking up on in order to separate "same" and "different" stimuli in these cases.

In the case of CLIP ResNet-50, the linear probe does not generalize to any OOD stimuli. On the other hand, CLIP ViT-B/16 probes trained on Shapes or Naturalistic stimuli generalize somewhat well to each other (90.4% generalization from Shapes to Naturalistic; 70.6% from Naturalistic to Shapes). Somewhat surprisingly, the CLIP ViT-B/16 probe trained on the Squiggles dataset does not generalize the relation to other datasets despite the impressive generalization performance of the fully fine-tuned model.

### A.8 CLIP Embedding Cosine Similarity Distributions

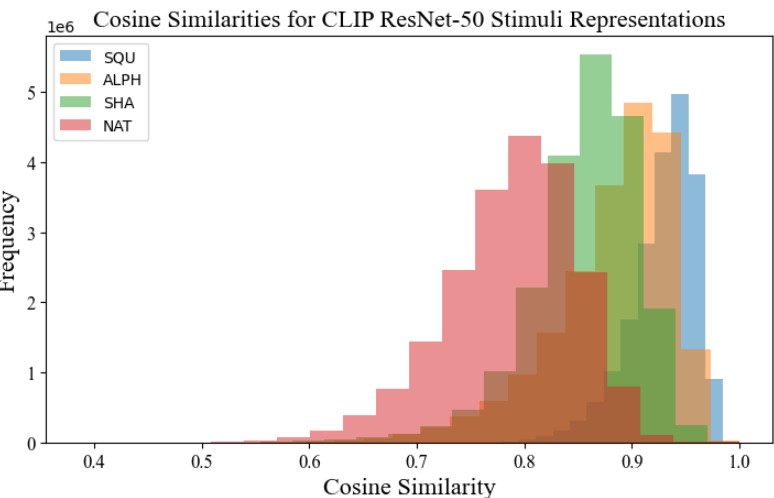

Figure 18: **Distribution of cosine similarities between CLIP ResNet-50 representations of the Squiggles, Alphanumeric, Shapes, and Naturalistic datasets.** These cosine similarities are calculated *before* fine-tuning. $n = 6,400$ for each dataset, 20.48M pairs calculated per dataset.

Table 14: **Average pairwise cosine similarity between CLIP embeddings of training stimuli within each dataset.** Because $n = 6,400$ for each dataset, averages are computed over 20.48M pairs. We extract CLIP embeddings *before* fine-tuning on the same-different task and *after* fine-tuning on the Squiggles task (median across five seeds).

| | *Before Fine-tuning* | | *Fine-tuned on SQU* | |
|---|---|---|---|---|
| Dataset ↓ | ResNet-50 | ViT-B/16 | ResNet-50 | ViT-B/16 |
| noise | 0.992 | 0.993 | 0.983 | 0.997 |
| SQU | 0.929 | 0.940 | 0.992 | 0.283 |
| ALPH | 0.881 | 0.889 | 0.984 | 0.634 |
| SHA | 0.855 | 0.861 | 0.949 | 0.548 |
| NAT | 0.788 | 0.805 | 0.937 | 0.568 |
| SHA-G | 0.868 | 0.873 | 0.938 | 0.538 |
| SHA-M | 0.900 | 0.904 | 0.948 | 0.579 |
| NAT-G | 0.845 | 0.850 | 0.944 | 0.513 |
| NAT-M | 0.882 | 0.879 | 0.940 | 0.407 |

Interestingly, Table 14 shows that ViT-B/16's embeddings seem to become more distinct during fine-tuning whereas ResNet-50's become closer together. This is likely *not* due to differences in generalization perfor-

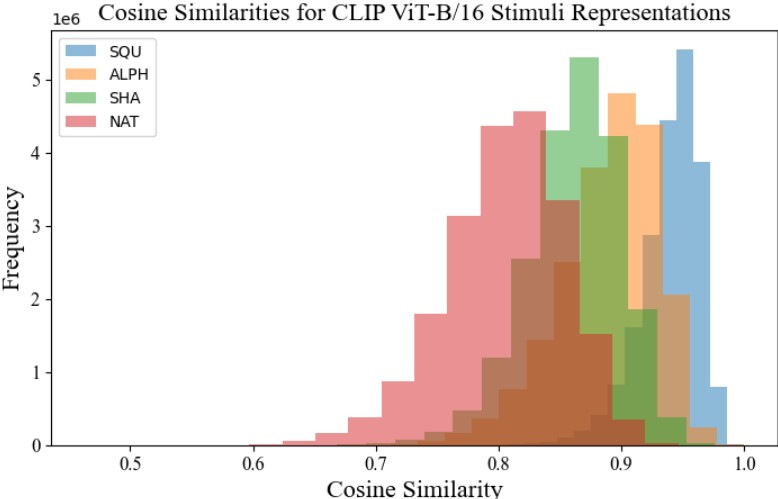

Figure 19: **Distribution of cosine similarities between CLIP ViT-B/16 representations of the Squiggles, Alphanumeric, Shapes, and Naturalistic datasets.** These cosine similarities are calculated *before* fine-tuning. $n = 6,400$ for each dataset, 20.48M pairs calculated per dataset.

mance given that the median difference between ViT-B/16 and ResNet-50 for within-distribution generalization is only 1.9%, and the median difference in out-of-distribution generalization is 13.9%. We do not have a clear explanation for this phenomenon, and also concede that it may be a methodological problem resulting from calculating cosine similarity between CLIP embeddings after extensive fine-tuning.

Given our hypothesis that generalization accuracy should correlate with greater cosine similarity of representations before fine-tuning, it is odd that the masked versions of Shapes and Naturalistic sometimes have greater average cosine similarity measures than Alphanumeric, despite having worse generalization accuracy (Tables 5-9). However, this is likely due to the fact that masking shapes greatly decreases the effective number of unique tokens in the dataset. For example, the Shapes dataset only has 16 unique shapes, so masking those objects results in only 16 unique objects in total. Appendix C shows that training on a dataset with so few tokens is detrimental to in- and out-of-distribution generalization. Thus, datasets with a high average cosine similarity seemingly only improve generalization in the cases where they also include a diversity of unique training objects (like the Squiggles dataset).

### A.9  Fine-tuning on Noise

We initially calculated average pairwise cosine similarity for CLIP representations of random Gaussian noise as a baseline for measuring visual diversity within our datasets (Table 2). However, after observing a pattern in which more closely-embedded datasets induce stronger out-of-distribution generalization, we decided to see whether models perform even better when they are fine-tuned on a version of the same-different task where they must label two same-versus-different 64x64 squares of random Gaussian noise (see Figure 20). Theoretically, if models fine-tuned on this task are forced to compare objects on the level of individual pixels, they should be able to generalize to any same-different dataset in which objects are the same on a pixel level (the definition of sameness we employ in this work).

We use the same methodology as described in Section 2. That is, we fine-tune CLIP ResNet-50 and CLIP ViT-B/16 on this task, sweeping over the learning rates (1e-4, 1e-5, 1e-6, 1e-7, 1e-8) and two learning rate schedulers (`Exponential`, `ReduceLROnPlateau`). We report results for the best models trained for 70 epochs with a batch size of 128 in Table 15.

As shown in Table 15, models fine-tuned on noise largely fail to generalize. One likely explanation for this lack of generalization is that models fine-tuned on noise learn to attend to small regions in both objects (e.g.

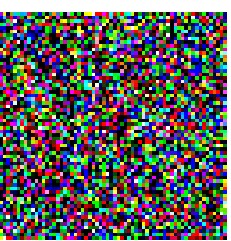 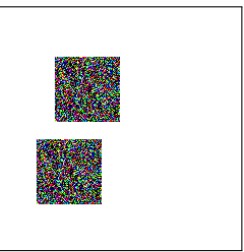 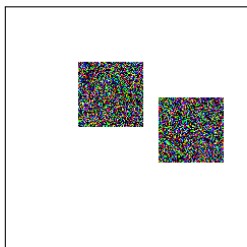

Figure 20: **Examples of stimuli used when fine-tuning on noise.** From left to right: a single example object; a stimulus labeled as "same;" a stimulus labeled as "different." All noise stimuli were sampled from a Gaussian distribution with $\mu = 0$ and $\sigma = 1$.

Table 15: **Out-of-distribution test accuracy for CLIP models fine-tuned on noise.** Rows indicate model architecture and number of epochs, while columns indicate the test dataset. Each cell is the median performance over five random seeds. The rightmost column labeled "Avg." is the row-wise average of accuracy scores across OOD evaluation sets (i.e. not including the NOISE column), which indicates how well a model is able to generalize to other datasets. The bottom row labeled "Avg." is the column-wise average, indicating how difficult it is for models fine-tuned on noise to generalize to that given dataset.

| | | | $\leftarrow$ **Test** $\rightarrow$ | | | |
| Model $\downarrow$ | NOISE | SQU | ALPH | SHA | NAT | Avg. |
|---|---|---|---|---|---|---|
| ViT-B/16 | **95.3** | 50.3 | 65.1 | 97.1 | 96.9 | 77.4 |
| ResNet-50 | **94.9** | 50 | 50 | 61.2 | 59.3 | 55.1 |
| Avg. | 95.1 | 50.2 | 57.6 | 79.2 | 78.1 | |

two adjacent pixels in the corner of each object) and calculate whether those small regions are equivalent. This might help explain why CLIP ViT-B/16 fine-tuned on noise generalizes quite strongly to the SHA and NAT datasets—these two datasets contain textures, so this potential strategy of computing equality based on highly localized features would work well. On the other hand, this strategy would likely fail for stimuli in the Squiggles and Alphanumeric tasks, which consist of primarily empty space and require the integration of more global shape information. Although the idea of training on noise for abstract-relations is promising in theory (since there should not be spurious, non-generalizing visual features), it would require careful design to counteract such undesirable local "shortcuts" (Geirhos et al., 2020).

## A.10 Sensitivity of OOD Generalization to Random Seed

In Table 1, we report median out-of-distribution test accuracy across five random seeds for CLIP ResNet-50 and CLIP ViT-B/16. Here, we extend this table by reporting out-of-distribution test accuracy for all five random seeds.

All model configurations demonstrate some sensitivity to random seed. However, the two best generalizing models—CLIP ResNet-50 fine-tuned on ALPH (Figure 21B) and CLIP ViT-B/16 fine-tuned on SQU (Figure 21E)—demonstrate a distinct bimodal distribution across seeds. While some seeds attain high test accuracy across all three OOD test sets, one (CLIP ViT) or two seeds (CLIP ResNet) perform substantially worse across all three sets. This creates a visible gap between points that persists across all three OOD test sets in panels B and E in Figure 21. Other configurations demonstrate such a gap for one or two test sets (e.g. panel A and panel D in Figure 21), but no other configurations demonstrate such a gap for all three OOD sets.

It is interesting to consider the fact that the only randomness in our setup for these models is in the data batching (since models are initialized with deterministic, pretrained weights). This indicates that the order in which models see particular examples from the training set is important for abstraction and determines whether or not models discover the generalizing solution.

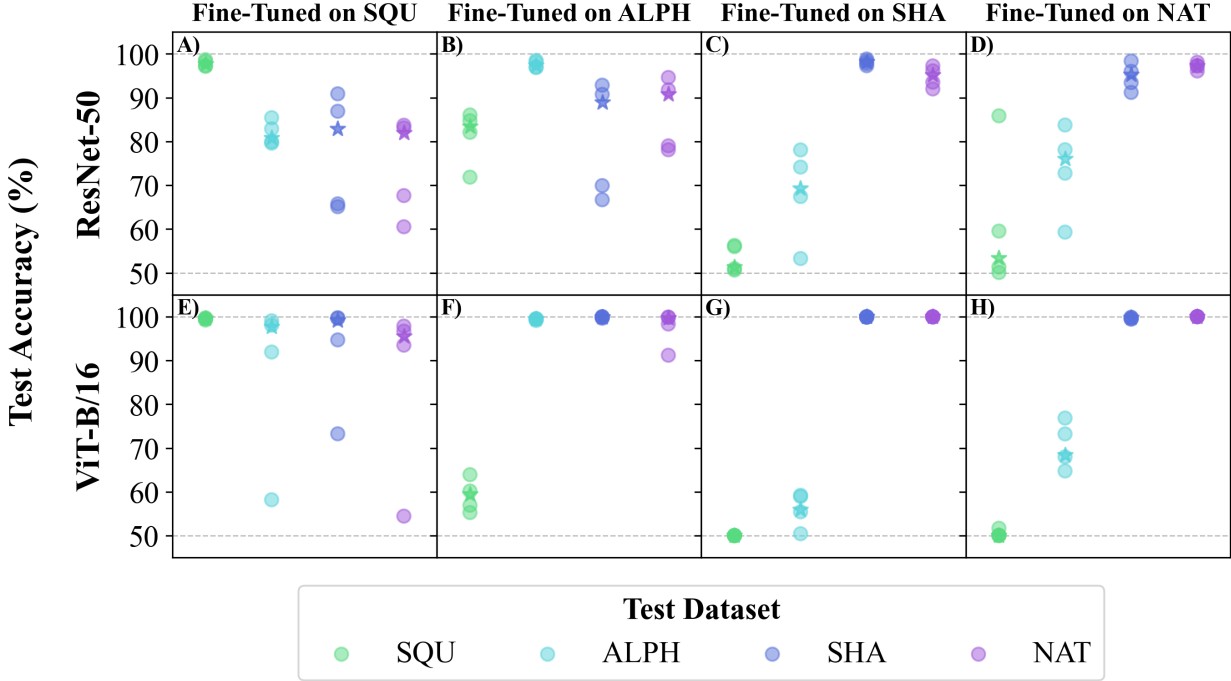

Figure 21: **Out-of-distribution test accuracy for CLIP models for each fine-tuning dataset across all five random seeds.** The top row shows test accuracy for CLIP ResNet-50, while the bottom row shows test accuracy for CLIP ViT-B/16. The columns indicate the fine-tuning dataset (from left to right: SQU, ALPH, SHA, & NAT), while the legend indicates the test dataset. Each individual plot point is the test accuracy for a given random seed. Stars represent the median test accuracy, which are equivalent to the values reported in Table 1.

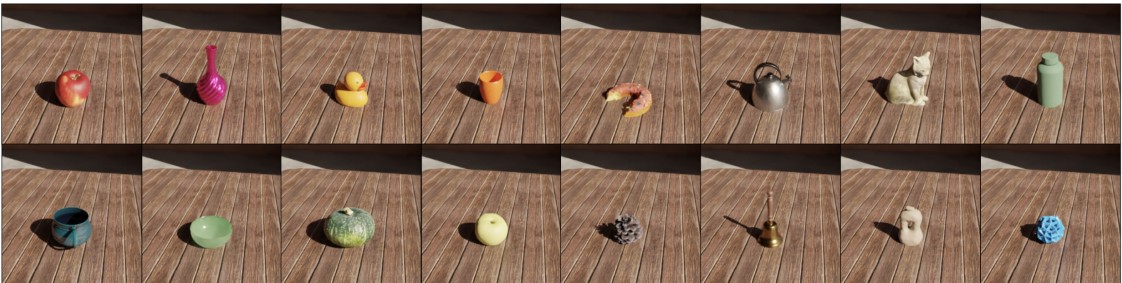

Figure 22: **Images of all 16 3D objects used to create the photorealistic evaluation set in Section 3.3.** Note that many of the objects lack rotational symmetry, e.g. the rubber duck (top row, third image) or the mug (bottom row, first image)—thus, different views of these objects can appear substantially different.

### A.11 Additional Photorealistic Evaluation Results

Using the objects depicted in Figure 22, we create two conditions of the photorealistic evaluation dataset described in Section 3.3: one in which individual objects in a given image are randomly and independently rotated, and one in which objects are given the same random rotation. The first condition presents a more challenging out-of-distribution task for our fine-tuned models than the second since it introduces additional and substantial variation between "same" objects.

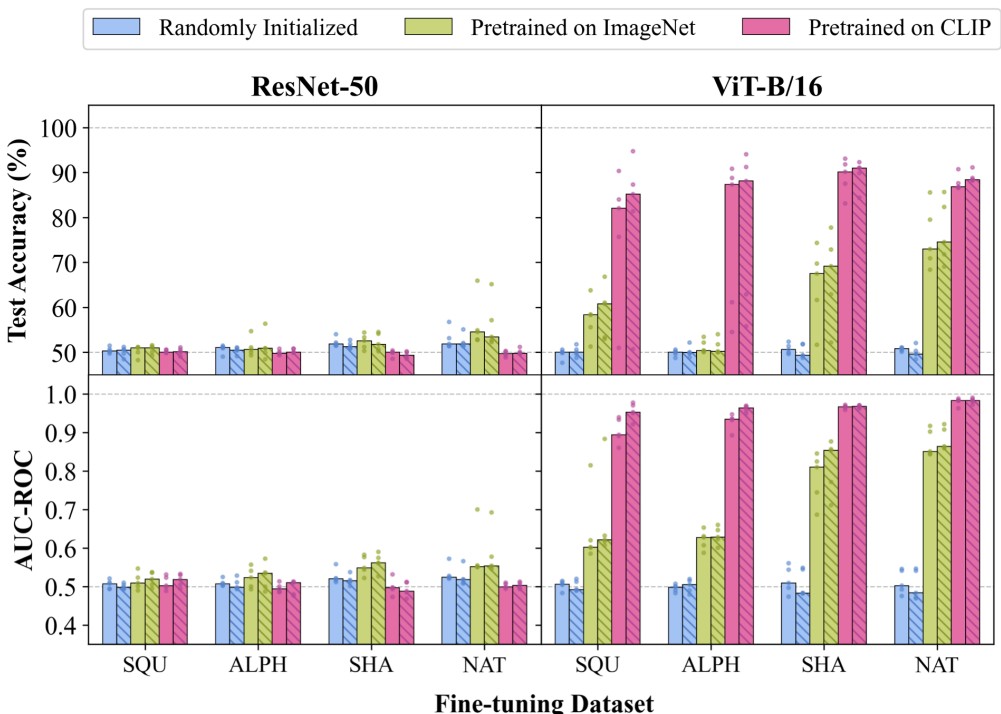

Figure 23: **Median test accuracy (top row) and AUC-ROC (bottom row) for models fine-tuned on SQU, ALPH, SHA, & NAT and tested on the photorealistic dataset.** The two plots on the left show results for ResNet models, while the two on the right show results for ViT. The bars are grouped by fine-tuning dataset, as indicated by the labels along the x-axis. The colors indicate the pretraining method. Hatched bars indicate model performance on the version of the photorealistic dataset in which objects are given identical random rotations; unhatched bars indicate model performance on the version in which individual objects are rotated independently. Individual seeds are also shown over each bar; these seeds are identical to those used in Sections 3.1 and 3.2.

We evaluate our SQU, ALPH, SHA, & NAT fine-tuned models as described in Section 3.3 on both conditions of the photorealistic dataset. None of the models receive any additional fine-tuning on the photorealistic dataset. Results for all pretraining and fine-tuning combinations are displayed in Figure 23—the hatched bars indicate the easier identical rotation condition, while unhatched bars indicate the more difficult individual rotation condition. CLIP ViT models demonstrate impressive generalization to the photorealistic stimuli across all fine-tuning datasets. ImageNet pretrained ViT models that are fine-tuned on the SHA and NAT datasets demonstrate some generalization to the photorealistic setting. All other models fail to generalize. In particular, although CLIP ResNet-50 demonstrates a similar generalization pattern to CLIP ViT-B/16 in Section 3.2 as shown in Table 1, none of the ResNet models generalize robustly to the photorealistic dataset. This suggests that ResNet models may be prone to relying on pixel-level heuristics.

Performance improves slightly for most models when objects are rotated identically. However, models perform nearly just as well when objects are rotated individually. This is impressive in the case of CLIP-pretrained ViT, seeing as models were not fine-tuned for rotational invariance. Evaluation results on the Lines dataset from Puebla & Bowers (2022) in Appendix A.1 seem to support the possibility that CLIP ViT models acquire a same-different relation that is also reflection invariant despite receiving no signal to do so.

# B    Inductive Bias Experiment Details

## B.1    Grayscale and Mask Details

Details on the training datasets are as follows:

**Grayscaled Shapes.** Images were taken from the Shapes dataset (Section 2) and converted to grayscale using the PIL `ImageOps.grayscale` method.

**Masked Shapes.** Images were taken from the Shapes dataset. Because the background was already white, we selected RGB pixels that were $\leq$ (250, 250, 250) and replaced them with pixels of the value (100, 100, 100). Extra pixels with any values greater than 250 that are not equal to the background color (255, 255, 255) were also converted to (100, 100, 100).

Because training datasets are constructed by sampling random objects, the exact objects used between the original, Grayscale, and Masked datasets are not the same.

## B.2    Dissociating Color, Texture, and Shape

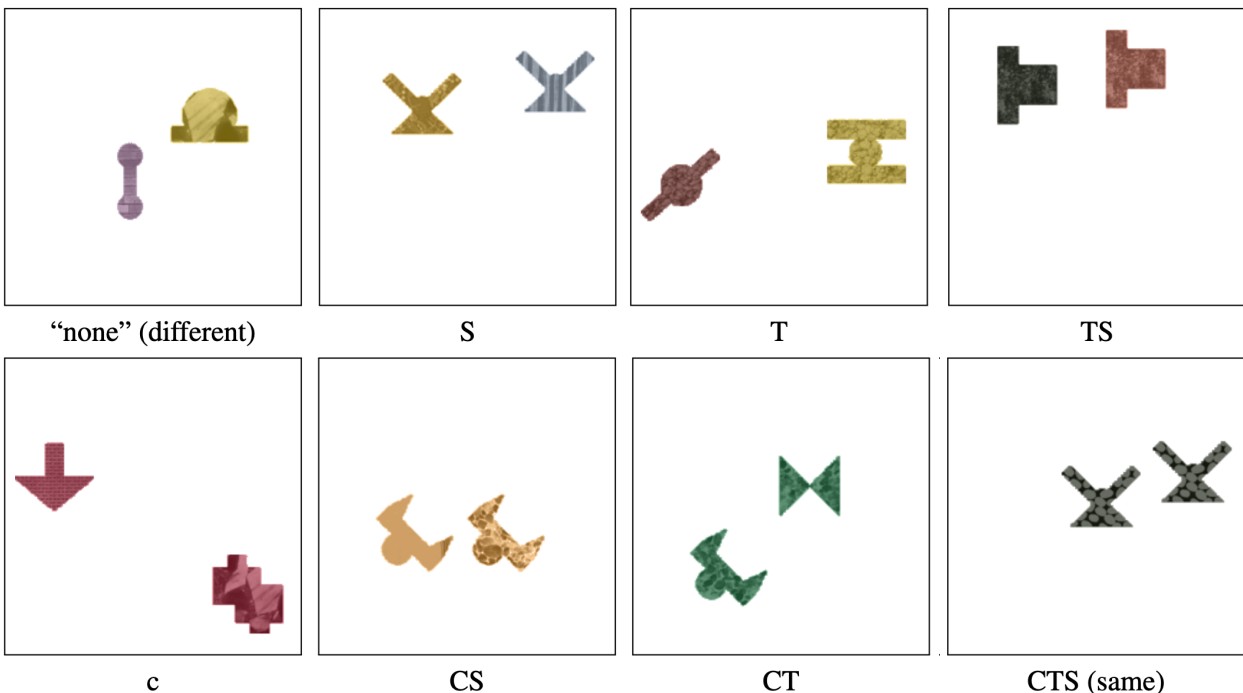

Figure 24: **Examples of training images from every Table 3 and Table 16 testing dataset.** Every test dataset contained 6400 images and 300 unique objects.

Table 16: **Predicted results of dissociation experiments, along with actual results from all models trained on different versions of the original Shapes dataset.** Ideally, the proportion of "same" predictions for different types of images should change based on the inductive bias a given model is using. Median results over five seeds are reported for each row. *SHA=Color Shapes, GRAY-SHA=Grayscale Shapes, MASK-SHA=Masked Shapes.*

| | *Acc.* | *Proportion of "Same" Predictions* | | | | | | | |
|---|---|---|---|---|---|---|---|---|---|
| **Predicted ↓** | acc. | none | S | T | TS | C | CS | CT | CTS |
| (no bias) | 1.00 | 0.00 | 0.00 | 0.00 | 0.00 | 0.00 | 0.00 | 0.00 | 1.00 |
| color | 1.00 | 0.00 | 0.00 | 0.00 | 0.00 | 1.00 | 1.00 | 1.00 | 1.00 |
| texture | 1.00 | 0.00 | 0.00 | 1.00 | 1.00 | 0.00 | 0.00 | 1.00 | 1.00 |
| shape | 1.00 | 0.00 | 1.00 | 0.00 | 1.00 | 0.00 | 1.00 | 0.00 | 1.00 |
| **ViT-B/16 ↓** | acc. | none | S | T | TS | C | CS | CT | CTS |
| SHA (Rand) | 0.91 | 0.15 | 0.15 | 0.17 | 0.16 | 0.86 | 0.87 | 0.96 | 0.97 |
| GRAY-SHA (Rand) | 0.77 | 0.33 | 0.35 | 0.45 | 0.50 | 0.41 | 0.48 | 0.80 | 0.87 |
| MASK-SHA (Rand) | 0.61 | 0.52 | 0.65 | 0.55 | 0.66 | 0.59 | 0.68 | 0.63 | 0.73 |
| SHA (ImageNet) | 1.00 | 0.00 | 0.02 | 0.01 | 0.06 | 0.34 | 0.81 | 0.82 | 1.00 |
| GRAY-SHA (ImageNet) | 1.00 | 0.00 | 0.01 | 0.00 | 0.06 | 0.05 | 0.40 | 0.47 | 1.00 |
| MASK-SHA (ImageNet) | 1.00 | 0.00 | 0.15 | 0.00 | 0.28 | 0.00 | 0.82 | 0.03 | 1.00 |
| SHA (CLIP) | 1.00 | 0.00 | 0.01 | 0.03 | 0.09 | 0.12 | 0.41 | 0.89 | 1.00 |
| GRAY-SHA (CLIP) | 1.00 | 0.00 | 0.00 | 0.01 | 0.06 | 0.02 | 0.26 | 0.59 | 1.00 |
| MASK-SHA (CLIP) | 1.00 | 0.00 | 0.04 | 0.00 | 0.24 | 0.00 | 0.47 | 0.02 | 1.00 |
| **ResNet-50 ↓** | acc. | none | S | T | TS | C | CS | CT | CTS |
| SHA (Rand) | 0.83 | 0.25 | 0.29 | 0.34 | 0.35 | 0.43 | 0.44 | 0.71 | 0.90 |
| GRAY-SHA (Rand) | 0.84 | 0.27 | 0.29 | 0.39 | 0.41 | 0.38 | 0.40 | 0.81 | 0.96 |
| MASK-SHA (R) | 0.79 | 0.26 | 0.36 | 0.37 | 0.48 | 0.34 | 0.47 | 0.47 | 0.85 |
| SHA (ImageNet) | 0.93 | 0.15 | 0.49 | 0.17 | 0.59 | 0.39 | 0.94 | 0.43 | 1.00 |
| GRAY-SHA (ImageNet) | 0.79 | 0.41 | 0.64 | 0.44 | 0.65 | 0.59 | 0.96 | 0.61 | 0.99 |
| MASK-SHA (ImageNet) | 0.84 | 0.17 | 0.49 | 0.17 | 0.45 | 0.27 | 0.83 | 0.28 | 0.85 |
| SHA (CLIP) | 0.98 | 0.04 | 0.11 | 0.05 | 0.15 | 0.20 | 0.60 | 0.47 | 1.00 |
| GRAY-SHA (CLIP) | 0.98 | 0.04 | 0.42 | 0.07 | 0.54 | 0.06 | 0.53 | 0.15 | 1.00 |
| MASK-SHA (CLIP) | 0.98 | 0.04 | 0.90 | 0.05 | 0.95 | 0.05 | 0.92 | 0.07 | 1.00 |

# C Diversity of Training Data Heatmaps

Figure 25: **Validation accuracies for a ViT-B/16 ImageNet model fine-tuned on different numbers of unique objects and different amounts of Squiggles stimuli.** Hyperparameters chosen correspond with the best-performing Squiggles model from Figure 3. Each cell is averaged over five different seeds. ImageNet ViT-B/16 must be fine-tuned on at least 25,600 images containing at least 1,024 unique tokens to achieve high out-of-distribution accuracy.

# D Patch Alignment Experiment

One intuition is that ViT models may be able to more easily compute same-different due to their ability to directly compare image patches using attention. This implies that if objects were aligned with ViT image patches, it might be easier for ViT models to implement the same-different relation (since segmentation would effectively already be done for the model).

We consider whether aligning objects with ViT patches allows for quicker convergence or more robust in-distribution generalization. Figure 26a and 26b show stimuli under the Aligned condition, where objects are aligned within the grid of tokens used by ViT models to process images. For ViT-B/16, each object takes up a 4x4 sub-grid of tokens (16 total); for ViT-B/32, objects take up 2x2 sub-grids (4 tokens total). The sub-grids in which the objects are placed are randomly chosen for each stimulus. The number of possible spatial configurations is exactly 36 for same stimuli (9 choose 2) and 72 for different stimuli.

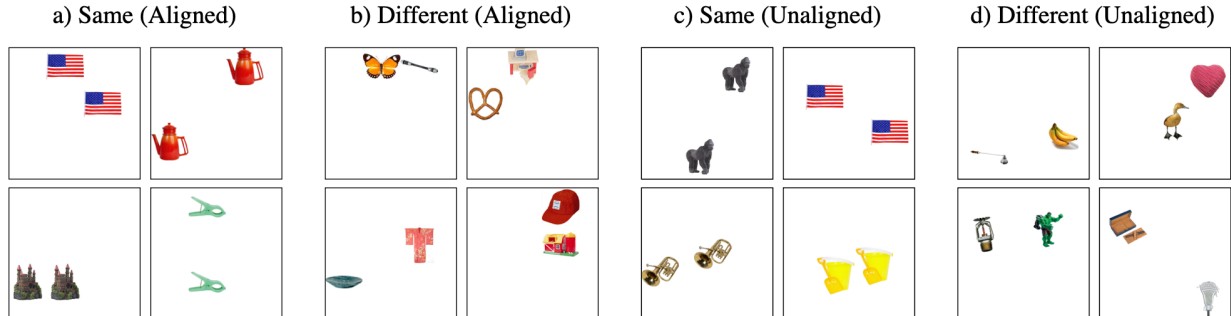

Figure 26: **Examples of stimuli in the Aligned condition (a) and (b) and the Unaligned condition (c) and (d).** Objects used are from the same Naturalistic dataset in the main paper (Brady et al., 2008).

Table 17: **Results for ImageNet models fine-tuned on stimuli from Figure 26.** Training accuracy, in-distribution validation accuracy, and out-of-distribution generalization to the Shapes dataset is shown.

| Aligned ↓ | Train Acc. | Val. Acc. | SHA Acc. |
|---|---|---|---|
| ViT-B/16 | 100 | 100 | 85.6 |
| ViT-B/32 | 100 | 99.7 | 82.4 |
| ResNet-50 | 87.2 | 68.9 | 53.9 |
| ResNet-152 | 99.5 | 89.2 | 74.6 |
| Unaligned ↓ | Train Acc. | Val. Acc. | SHA Acc. |
| ViT-B/16 | 100 | 100 | 91.0 |
| ViT-B/32 | 100 | 99.5 | 96.9 |
| ResNet-50 | 85.7 | 66.9 | 55.9 |
| ResNet-152 | 99.5 | 88.6 | 78.5 |

On the other hand, Figure 26c and 26d show stimuli under the Unaligned condition. In this case, stimuli are randomly placed and do not have to align with ViT tokens (just as in the rest of our experiments). The result is that the objects span a larger number of tokens, and the number of configurations that the objects can occupy from the point of view of the ViT is combinatorially much larger than in the Aligned condition. Thus, ViT models trained on these stimuli must integrate information across a larger and much less predictable set of tokens. The number of possible spatial configurations is on the order of 100 million.

In all experiments, models are trained to classify images as same or different with cross entropy loss and a batch size of 64 for 30 epochs. Each experiment uses an initial learning rate of 2e-6, a ReduceLROnPlateau learning rate scheduler (patience=2), and an AdamW optimizer (weight decay=1e-2). Models are fine-tuned on 6400 stimuli (with 1920 unique training objects, disjoint from 240 other validation objects). [6]

Table 17 shows results for ImageNet models fine-tuned on these datasets. Contrary to our hypothesis, it seems that ViT models do not benefit from having objects aligned to their token patches. In fact, the Unaligned condition provides slightly better generalization, likely because there is more variability in the training data.

---

[6]The setup of this experiment is slightly different from the main paper is because it was an early exploratory result.

