# OpenReview forum: "Deep Neural Networks Can Learn Generalizable Same-Different Visual Relations"
_TMLR — Rejected by TMLR_

### Review · Reviewer_1Dkn · 2024-07-13

**Summary Of Contributions:**

This work aims to test whether a combination of pretraining regime and model architecture can yield a neural network model capable of learning abstract generalizable same-different visual relations. To my knowledge this is the first work testing whether CLIP pretrained ViTs on visual relational reasoning.

**Audience:**

Yes

**Broader Impact Concerns:**

None.

**Claims And Evidence:**

No

**Requested Changes:**

- Since the main claims of this paper directly contradict the ones of Puebla and Bowers (2022), I think the authors should test the full set of 13 OOD test datasets of Puebla and Bowers (2022) and move the results to the main body of the paper instead of the appendix. Depending on their results, the authors should revaluate their claims regarding the capacity of the CLIP pretrained ViT model to learn abstract relations.

- The authors should acknowledge the change in task definition in section 3.3 and discuss the limitations in doing so for the interpretations of the results of this section.

**Strengths And Weaknesses:**

Strengths:

- The authors present simulations in a well designed new dataset.
- The analysis are very detailed.

Weaknesses:

- The authors state that a model that learned the same/different relation should generalize its discrimination to any new image. However, in their tests of Puebla and Bowers (2022) they found the ViT model failed in two datasets. Furthermore, they left out the four additional datasets used in Simulation 5 of Puebla and Bowers (2022). Puebla and Bowers (2024) trained in the dataset of Fleuret et al. (2011) and found that the CLIP pretrained ViT did not generalized to these additional datasets either.

- In section 3.3 the authors test OOD generalization in 3D photorealistic stimuli, but in doing so they changed the definition of the same/different task. As the authors correctly point out, the definition used in most previous research defines same as “exactly the same up to translation in the canvas”. The authors use the fact that the CLIP pretrained ViT model generalizes to the photorealistic setting to argue that the model has learned a “more abstract” notion of sameness, but other interpretations are equally plausible. For example, it could be that the model is exhaustively comparing patch embeddings and deciding according to an internal similarity threshold. Note that in this hypothetical case what is being compared are not objects but locations on the canvas. In part because this simulation is using a task definition different from the original one is hard to decide between different explanations.

References not in the manuscript:
- Puebla, G., & Bowers, J. S. (2024). Visual Reasoning in Object-Centric Deep Neural Networks: A Comparative Cognition Approach. arXiv preprint arXiv:2402.12675.

---

> ### Author Response · Authors · 2024-08-02
>
> Thank you for your time. To respond to your points,
> - You’re right that CLIP ViT did not generalize to two of the Puebla and Bowers datasets, but there are particular reasons for this gap. The dataset that CLIP ViT performed worst on consisted of a single “S” shape reused throughout all images, where images labeled “different” consist of two occurrences of the identical shape with one vertically flipped. Therefore, the “different” objects are actually the same under reflection—the “failure” of CLIP ViT on this dataset is a case of mistaking a backwards “S” for a forwards “S” (all “different” images are classified as “same” with high confidence), which we believe to be an exception that proves the robustness of the model. For the other dataset, we show examples of images that were classified incorrectly by the model, finding that the objects are frequently so close together that they almost overlap; objects in the training images are never this close, so models may mistakenly judge that there are less than two objects in the image.
> - To clarify, when we claim that success on the 3D object task implies an “abstract” notion of same-different, we simply mean that the model has learned to make comparisons between higher-level visual features (rather than pixel-level information). We find it surprising that a model fine-tuned to classify 2D images in which objects are either pixel-level same or different performs almost as well when tested on 3D stimuli, in which objects vary at the pixel level even for “same” pairs. The model could have easily learned any number of less general approaches that rely on extremely specific low-level visual features. It seems plausible that at an algorithmic level, any model (including CLIP ViT) would have to exhaustively compare patch tokens to solve this task. But if the model learns a solution that generalizes to an extremely OOD dataset (especially the 3D dataset, in which object patches are never identical on the pixel-level), we claim it is justifiable to describe that approach as “more abstract” on a high level.
> - Thank you for the pointer to the new preprint! We will be sure to read and consider it in future work.

---

> ### Comment · Reviewer_1Dkn · 2024-08-07
>
> Thank you for your replies. I'll comment on them point by point:
>
> - Regarding the performance on the Arrows dataset. Looking at the examples given in Figure 13 the arrows do not seem to be specially close. In any case, the process of recognizing a relation between objects from raw pixels entails segregating the objects from the background and isolating each individual object to compare them. A failure in object segregation is a failure in relation recognition, specially if the relations are easily recognizable by a human observer.
>
> - Regarding the performance on the Lines dataset. The explanation given for the model's low performance is post hoc and there are other explanations as likely. For example, it can be that the model has problems dealing with the configural aspects of an obtect's shape (this is also consistent with the slight decrease in performance in the Scrambled dataset).
>
> - Testing the CLIP ViT model in the remainder four additional datasets of Puebla and Bowers (2022) is a minimal request since all their datasets are freely available in the same code repository. I don't think is acceptable to skip datasets because their are inconvenient for the paper's conclusions (nor it is to put results inconsistent with the paper's conclusions in the appendix).
>
> - The word abstract has a specific meaning: "thought of apart from concrete realities or specific objects". This is the sense of the word used throughout the paper (an abstract relation should generalize broadly because is not tied to particular instances). Re-defining the meaning of the word to "comparing higher-level visual features" is therefore inappropriate. Furthermore, this point does not address the fact that the task definition of 3.3 is different from the definition used elsewhere in the paper.

---

> ### Author Response · Authors · 2024-08-15
> **Updated Manuscript Addressing Concerns**
>
> _(**Note**: A specific point-by-point response is in the comment below titled "Response to Specific Points." This comment addresses the broader themes in Reviewer 1Dkn's reply and enumerates important updates to our manuscript.)_
>
> \
> **General Response & Updates to Our Manuscript**:
>
> Thank you for your responses—your points are well taken, and your concerns regarding the limitations of CLIP ViT are valid. We believe that since CLIP ViT achieves excellent or near perfect performance on most of the OOD datasets tested, the model has learned a nontrivial, generalizable solution that abstracts away from low-level visual features. However, The model’s difficulties on the Arrows and Lines datasets are important results, and we did not intend to deemphasize or hide them. We have since **1)** added a pointer to this section in the introduction, **2)** expanded our discussion on this section in the discussion, **3)** reworded some of our claims in the main text regarding “genuine” abstraction, favoring a "fuzzy" relation interpretation of our results instead (see below), **4)** reorganized the Appendix such that the Puebla & Bowers (2022) evaluation results are the first section (A.1), and **5)** added evaluations on the four additional datasets from Puebla and Bowers (2022) to A.1.
>
> \
> **Four Additional Evaluation Sets from Puebla & Bowers (2022)**:
>
> We evaluate CLIP ViT fine-tuned on SQU on Rectangles, Straight Lines, Connected Squares, & Connected Circles from Puebla & Bowers (2022) as requested. The model struggles to generalize strongly to them. The new median ROC-AUC scores are the following:
> - Rectangles: 0.6291
> - Straight Lines: 0.5643
> - Connected Squares: 0.5049
> - Connected Circles: 0.7568
>
> These results are now in Appendix A.1.1 (also see Figure 11 in the updated manuscript).
>
> \
> **CLIP ViT Appears to Learn a "Fuzzy" Same-Different Relation**:
>
> We do not believe that the results on the four additional evaluation sets (above) necessarily contradict our conclusions. The fact that CLIP ViT generalizes strongly to the photorealistic dataset in Section 3.3 suggests that the model has learned a **“fuzzy” relation**: rather than judging same vs. different using exact equality between low-level visual features or pixels, the model is likely using a threshold of similarity to determine equality between objects. Therefore, if objects in an OOD stimulus are considered “similar enough” in embedding space by the model, they will be judged “same” even if they are pixel-level different. We have arrived at this explanation after conducting **several additional investigations** using the datasets from Puebla and Bowers (2022) as well as our own datasets:
> 1. CLIP ViT indeed can discriminate between “same” and “different” images for Rectangles, Straight Lines, and Connected Circles to some extent: the average “same” logit assigned to “different” images is reliably lower than the average “same” logit assigned to true “same” images (3.45 vs. 3.82 for Rectangles; 3.61 vs. 3.81 for Straight Lines; 3.2 vs. 4.0 for Connected Circles). See Appendix A.1.1 and Table 4 in the updated manuscript.
> 2. We embed individual objects from each dataset using the image encoder of CLIP ViT fine-tuned on SQU, then compute the average pairwise cosine similarity between embeddings of “different” objects. We find that the average similarity between objects of a given dataset is strongly correlated with model generalization to that dataset: model performance suffers on datasets with higher similarity than the fine-tuning dataset (SQU). (In particular, this includes Lines, Rectangles, Straight Lines, Connected Squares, and Connected Circles). See Appendix A.1.2 and Figure 12 in the updated manuscript.
> 3. CLIP ViT errors on datasets with highly similar objects (see above point) demonstrate a clear pattern: the model predicts nearly all images as “same,” further suggesting that the model is using a similarity threshold to judge same vs. different. In particular: 99.97% of all stimuli are predicted “same” for the Rectangles dataset, 99.98% for the Straight Lines dataset, 100% for Connected Squares, and 99.95% for Connected Circles. See Table 4 in the updated manuscript.
>
> We thus argue that model performance on Arrows, Lines, and the additional datasets does not indicate failure to learn the relation per se; rather, the model fails on these additional datasets due to the relatively lower threshold of “sameness” it learns on its fine-tuning dataset. In fact, given that the model has been pretrained on real images and that objects in the real world are never exactly the same, this threshold solution can be considered more desirable than an exact pixel-level comparison depending on your use case.

---

> ### Author Response · Authors · 2024-08-15
> **Responding to Specific Points**
>
> **A Response to the Specific Points Raised**:
>
> Responding point-by-point:
> - We agree that the Arrows dataset is an interesting case. We have added some additional speculation to Appendix A.1 about the model’s difficulties with this dataset. It presents a unique challenge because oftentimes the only difference between two objects is where a line connects to an identically-oriented triangle. It is plausible that the model fails to properly represent these spatial differences between different objects. However, we consider “perceptual” failures such as this to be different from abject failures to learn the same-different relation; the issue is not necessarily that the model cannot apply the same-different relation, but that it fails to apply a specific spatial relation (that it is not trained for) at an earlier stage of its computation, resulting in lower quality object representations. The ability to form human-aligned object representations with neural networks is not at all trivial and should be considered separately from pure relational learning.
> - To test our intuition that CLIP ViT is reflection invariant, we construct a “flipped” version of the SQU, ALPH, & NAT datasets where “different” images are actually the same object but reflected as in the Lines dataset (see Figure 13 in the updated manuscript). We find that model performance for CLIP ViT fine-tuned on SQU (not flipped) suffers on these flipped datasets due to the model predicting most of the “different” images as “same.” Median test accuracy decreases from 99.6% to 77.1% for SQU (72.9% of all stimuli predicted “same”), 97.7% to 51.6% for ALPH (98.4% of all stimuli predicted “same”), and 96.7% to 51.8% for NAT (98.2% of all stimuli predicted “same”). CLIP ViT reliably considers objects to be the same under reflection, strengthening our explanation for performance on Lines. These results are now included in Appendix A.1.3. However, we do acknowledge that there could be other explanations, and we explicitly state our uncertainty in Appendix A.1.
> - We have added these results (Appendix A.1.1 and Figure 11 in the updated manuscript). We do not include these results in the main body of the paper due to the page limit, but we have added more conspicuous references to Appendix A.1 more generally (e.g. in the bulleted list of contributions in the introduction, a new paragraph in the discussion). We hope that these additional pointers will guide readers to the Appendix. We have also reorganized the Appendix such that this section appears first.
> - We believe that our definitions of “abstract” are actually in agreement with each other! In the case of a vision model, we understand the “concrete realities” or “specific objects” that the model operates on to be the pixel values of the input image. The high-level features we refer to are “abstract” by virtue of being removed by multiple stages of computation (layers) from the concrete, pixel-level properties of the input.

---

### Review · Reviewer_3AdN · 2024-07-15

**Summary Of Contributions:**

- The authors find that “_different fine-tuning datasets lead to qualitatively different patterns of generalization—fine-tuning on more visually abstract objects (which do not contain color or texture) results in stronger out-of-distribution generalization, whereas fine-tuning on more naturalistic objects fails to generalize_”

- The authors observe that “_ViTs generally prefer to determine equality between objects by comparing their color or texture, only learning to compare shape when the fine-tuning dataset lacks color and texture information. However, we find that CLIP pretraining helps to mitigate this preference for color and texture_”

**Audience:**

Yes

**Broader Impact Concerns:**

The manuscript does not have a Broader Impact Statement but I don't think it's necessary. I don't see any broader impact concerns with this work.

**Claims And Evidence:**

No

**Requested Changes:**

In its current form, I don’t feel comfortable accepting this manuscript to TMLR. This manuscript feels like work in progress and better suited for a workshop than for TMLR. A number of claims are too bold and not backed up by clear evidence. If this should be considered for publication at TMLR (which can potentially lead to a presentation at one of the top-tier ML conferences), then there are a number of changes that need to be made:

- Comparing an ImageNet-trained ViT with an ImageNet-trained ResNet-50 is not fair for several reasons: First, the people that trained those models are different and thus the hyperparameters and training recipes that these people used are different. Second, the training recipe that the people behind the ImageNet-trained ViT used are substantially newer than the training recipe that the people behind the ResNet-50 used. Third, the hyperparameter grid search that was used to train the ViT-B was probably much larger due to more compute. If you want to make a fair comparison between an ImageNet-trained ViT and an ImageNet-trained ResNet-50 you have to train those models on ImageNet yourself. I would not have requested such an experiment a couple of years ago, but in 2024 training models on ImageNet does not require excessively high compute, not even for academic labs. Any ML lab should be able to train reasonable sized models – of which a ViT-B and a ResNet-50 are ones – on ImageNet.

- Comparing a single CNN with a single ViT is really *not* a representative comparison. I would like to see a much larger pool of CNN-based models being compared to ViTs bevor I believe the following statement: “ViTs may perform the best on the same-different task because of their larger receptive field size; CNNs can only compare distant image patches in deeper layers, whereas ViTs can compare any image patch to any other as early as the first self-attention layer. Thus, ViTs may be able to integrate complex shape information and compare individual objects to each other more efficiently than CNNs.”

I mentioned the ConNext paper above. I am interested to see how ConvNext compares to ViT on the tasks that you use in your analyses. I am also interested in a couple of other CNN-based models. Again, please train ImageNet models yourself because only then it is guaranteed to yield a fair comparison or use ImageNet models only from the timm library and make sure that you use the most recent set of weights (with the best performance on ImageNet) that is available). I would not request this for CLIP pretraining because that’s simply not feasible but I think requesting this for ImageNet is fine.

- If you want to make the claim that you “_[…] compared a range of pretraining styles [...]_” and “_[…] thoroughly investigate the ability of neural networks to learn and generalize the same-different relation [...]_”, then please compare at the very least a few other (open-source) models and pretraining datasets. There’s the dataset that DINOv2 was trained on (which is different from ImageNet and CLIP), there is SWAG on which ViTs are trained (publicly available via torchvision), there is Laion-{400m,2B,5B}, etc. All of it is publicly available. Just have a look at the timm library, torchvision, and huggingface. I will not request training models from scratch on SWAG or any of the Laion datasets yourself. That is not feasible (whereas training on ImageNet clearly is) but I will request that you analyze models trained on these datasets and do a more thorough investigation.

**Strengths And Weaknesses:**

**Strenghts**:

- The research question of “_how neural networks encode and represent the same-different relation of objects_” is interesting and worth exploring.

- I like that the authors looked a randomly initialized networks which is a useful way to pin down the influence that inductive biases of different architecture classes have on a target task.

**Weaknesses**:

- In the introduction the authors write “_[…] recent work has argued
that deep neural networks struggle to learn this simple relation […]_”

First, lots of the cited work is not recent in the space of ML. All of the cited work that is pre-2020 is not recent. Second, if we look at the most recent works in the LLM world, then I don’t think that this motivation holds. GPT-4o and Gemini both seem to be able to learn this relation (for more details see the technical reports of GPT-4o and Gemini) and also DINOv2, SigLIP, or EVA-CLIP (have a look at the DINOv2, the SigLIP and EVA-CLIP papers).

- I don’t understand the last two sentences of the introduction “_Finally, most prior work focuses on convolutional architectures, but Vision Transformers (ViTs) (Dosovitskiy et al., 2020) adapted from the language domain (Vaswani et al., 2017) have recently emerged as a competitive alternative to CNNs on visual tasks. Self-attention, a key feature of ViTs, may provide an advantage when learning abstract visual relations—indeed, the ability to attend to and relate any part of a stimulus to any other part may be crucial for relational abstraction_”

What does “_adapted from the language domain_” mean? Also, did you read the ConvNext paper “A convnet for the 2020s” by Liu et al., 2022? ConvNext models are competitive with ViTs in a number of computer vision tasks. If you look at Table 1 and Table 2 in that paper, you can see that ConvNext models are competitive or even better than ViTs in almost every metric that is important for modern computer vision.

Why do you think that ViT “_may provide an advantage when learning abstract visual relations_” over CNNs? A 1D convolution can be seen as a linear layer and, vice versa, a linear layer (which most parts of a Transformer are composed of) can be regarded as a 1D convolution. I agree that the inductive biases between CNNs and ViTs are different but I disagree with the statement that a ViT is necessary for getting abstract relations right. The main advantage of a ViT is that it scales better than a CNN. Convolutions relate different parts of an image to one another and can thus probably be regarded as similar to what attention does by using image patches. There also exist CNNs with attention mechanisms.


- It is not surprising to me that CLIP pretraining yields the highest “in-distribution test set accuracy” even without fine-tuning (and just linear probing). Most of these shapes and their relations are probably in the training data used for CLIP pretraining and in addition text is used that contains words like “shape”, “different”, etc and aligned with the image encoder representations.

- How do you explain the “randomly initialized” results shown in Figure 3? In two out of four datasets it seems as if the inductive bias of a ResNet-50 is more useful for capturing the relationships than the inductive bias of a ViT-B/16. Only in one out of four datasets ViT-B/16 shows stronger performance.

- Similar to above, if we look at Figure 7, then there is no big difference between a ResNet-50 and a ViT-B/16. If any, then the ResNet-50 seems to have a better inductive bias for generalization to the Shapes dataset.

- In the discussion, the authors write “_In this article, we explored a range of architectures, pretraining styles, and fine-tuning datasets in order to thoroughly investigate the ability of neural networks to learn and generalize the same-different relation._” This is clearly a stretch. Which range of architectures and pretraining styles did you compare? I see a single CNN-based model compared against a single ViT-based model. This is not a range. These are two models. You compare publicly available ImageNet training against publicly available CLIP pretraining. Again, this is not a range. These are two pretraining styles for the same two models, which is pretty low effort. There are various other publicly available datasets out there. I don’t think that the ability of neural networks to learn and generalized the same-different relation has been thoroughly investigated here.

- The motivation says that ViTs are better at capturing the same-different relations than CNNs due to their different inductive biases. From the analysis regarding randomly initialized networks, it seems as if CNNs are better than ViTs in most cases. The randomly initialized analysis is the only comparison that is fair IMO. How do you explain this? As I mentioned elsewhere, I don’t think that comparing an ImageNet-trained ResNet-50 to an ImageNet-trained ViT is fair if you did not train the models yourself or that you can guarantee that the same tricks and training recipes were used. Deep Learning progress is incredibly fast, and training recipes and tricks become much more sophisticated and in addition more compute allows for more exhaustive grid searches.

---

> ### Author Response · Authors · 2024-08-02
>
> Thank you for your review and your time. To address your concerns,
> - Although it is possible that GPT-4o and Gemini are able to complete the same-different task, we were unable to find previous work explicitly testing this ability in either of these models’ technical reports, and it would be quite expensive to conduct a robust evaluation ourselves. Regardless of how well GPT-4o and Gemini can do on this task, it seems unlikely that a relation as simple as “same-different” would only be solvable by state-of-the-art language/vision models. Of course, we may be wrong: our only successful model was pre-trained with CLIP, which points to the possibility that NNs really do require large-scale pretraining with linguistic supervision to learn abstract relations (in which case GPT-4o and Gemini fit that mold). Nonetheless, we do not believe that a discussion of how NNs process abstract relations should start and end with large foundation models, especially when it comes to simple relations like same-different.
> - To address your point that the models we use are outdated, we add ImageNet-22k ConvNeXt results to Appendix A.3. ConvNeXt-B has 89M parameters, slightly more than ViT-B/16, which has 84M. Additionally, ConvNeXt was released about a year after ViT-B/16. Despite these advantages, it does not seem to generalize OOD as well as ViT in general (but is still strong compared to ResNet-50). We also compare ImageNet-1k DeiT-S, a small transformer model with 22M parameters, to ResNet-50 and find that DeiT-S generalizes more robustly.
> - You are correct to say that the two models we tested are not a wide range. We have revised the conclusion to more accurately summarize that we have proven the *existence* of models that are able to solve the same-different task without explicit guidance.
> - To address your comment about Figure 3, you are right to say that randomly-initialized ResNet-50 seems to outperform randomly-initialized ViT-B/16. Our assertion that ViTs are better at solving the same-different task may not be correct in cases without extensive pre-training. The gap between randomly-initialized ResNet and ViT, however, may also be found in the fact that ViT models are more data-hungry, requiring more pre-training data to develop the inductive biases already present in convolutional models (Dosovitskiy et al., 2021).
> - By “adapted from the language domain,” we are simply referring to the fact that transformer models were originally developed for machine translation before being applied to vision tasks.

---

### Review · Reviewer_muYt · 2024-07-17

**Summary Of Contributions:**

This works explores how would cnn and vit model would learn and generalize the abstract same-versus-different tasks (pixel-wise exact sameness) under different pretraining methodologies such as random initialization, pretraining, and contrastive multimodal pretraining (CLIP).

**Audience:**

Yes

**Broader Impact Concerns:**

I think the broader impact is sufficiently addressed.

**Claims And Evidence:**

Yes

**Requested Changes:**

- I would recommend to include extra architecture such as convnext, densenet, and vit-small and vit-tiny (e.g. the number of parameters for a cnn-based arch and vit-based arch are close) to explore if the findings/conclusions made in this paper are actually contributed by the training methodologies and model architectures rather than different number of parameters. Maybe including clip with vit-large for example in the comparison would also be nice. This would strengthen the work and findings from this work a lot in my view.

**Strengths And Weaknesses:**

Strength:
 The paper is well written. The experiments and datasets are conducted and constructed with careful thoughts. The overall findings are interesting.

Weakness:
My biggest concern comes from the fact that the parameter sizes vary a lot for the selected models. ResNet 50 has roughly 23 M parameters, and ViT-base has roughly 86 M parameters.
It is unclear if the findings from this paper such as the statement "ViTs may be able to integrate complex shape information and compare individual objects to each other more efficiently than CNNs" is because of the different sizes in parameters rather than model architecture itself. However, addressing such confounding variable seems to be missing in the study.

But picking resnet50 and vit-base is understandable as clip only has these as the smallest sized pretrained models.

---

> ### Author Response · Authors · 2024-08-02
>
> Thank you for your review!
> - You raise a good point about the mismatched parameter counts for ResNet-50 and ViT-B/16. To provide a fairer comparison, we have added results for ImageNet-22k ConvNeXt-B in Appendix A.3, as well as DeiT-Small.
> - ConvNeXt-B has 89M parameters, slightly more than ViT-B/16, which has 84M. Despite the fact that ConvNeXt-B is bigger than ViT-B, it does not generalize OOD as well as ViT. However, it slightly outperforms ImageNet-1k ResNet-50, which could either be due to the larger training dataset or the greater number of parameters in ConvNeXt.
> - We also provide results for a fine-tuned DeiT-B/16, which is a differently-trained model of the same size and architecture as ViT-base. Interestingly, although DeiT was only pretrained on ImageNet-1k, its OOD generalization is still comparable to ConvNeXt (trained on ImageNet-22k).

---

### Decision · Action_Editor_fyyH · 2024-09-16

**Recommendation:** Reject

**Comment:**

Reviewers praised the interest of this paper to the community, but raised concerns about the empirical evidence presented and the scope of claims made.

**Reviewer muYt** suggested to evaluate more models to isolate the effect of model dimensionality on performance on the same-different task. The authors added several more models in a revision, but the reviewer maintained a recommendation to incorporate a more extensive empirical evaluation in the main body of the paper. I assess that such an evaluation would help to substantiate the claim that self-attention is critical (or helpful).

**Reviewer 3AdN** also requested the evaluation of more models, as well as a tempering of the broad claims made in the submission. The authors addressed this by weakening claims to be about the existence of abstract visual reasoning in some cases.
The reviewer also suggested to demonstrate that training recipe is not a factor in performance on the same-different task by retraining models from scratch or by using a standardized library. The authors did not directly address this concern.

**Reviewer 1Dkn** evaluated that the mechanistic explanations for failures on 4 evaluation sets from Puebla & Bowers (2022) added in the revision are insufficiently evidenced, stating that:
> ... visual relation learning is a complex problem that involves a series of other subprocesses such as object segregation and spatial configuration recognition. A failure in any of these subprocesses can lead to a failure in the final relation recognition task. By accepting ad hoc explanations of failures in these subprocesses, one ends up changing the nature of the phenomenon studied.

I suggest to the authors to either temper the claim on the ability of CLIP pre-training to instill the ability to learn abstract visual relations, or otherwise substantiate the attested mechanistic explanation for these failures in a way that refutes challenges to this claim.

The authors' revision adjusted the claims made in the submission and provided more model evaluations as evidence, both improvements according to the reviewers' assessment. However, reviewers maintained substantial concerns after the discussion period about the degree of these improvements. The authors may consider submitting a revision to TMLR further addressing the reviewer's concerns.

**Note**: The authors provided changes as part of a submission revision but did not engage with reviewers until after the two-week discussion period had elapsed. Thus, some author responses were not seen by reviewers (namely Reviewer 3AdN). Nevertheless, I have taken the author response into account above.

**Audience:**

Yes, there are many in TMLR's audience interested in abstract visual reasoning.

**Claims And Evidence:**

Not entirely. In particular, reviewers had concerns related to both the empirical evidence and the scope of claims, detailed below.

**Resubmission Of Major Revision:**

The authors may consider submitting a major revision at a later time.